# Rapid response of fly populations to gene dosage across development and generations

Xueying C. Li [1,4] ✉, Lautaro Gandara [1], Måns Ekelöf [1], Kerstin Richter [1], Theodore Alexandrov [1,2,3] & Justin Crocker [1] ✉

Although the effects of genetic and environmental perturbations on multicellular organisms are rarely restricted to single phenotypic layers, our current understanding of how developmental programs react to these challenges remains limited. Here, we have examined the phenotypic consequences of disturbing the *bicoid* regulatory network in early *Drosophila* embryos. We generated flies with two extra copies of *bicoid*, which causes a posterior shift of the network's regulatory outputs and a decrease in fitness. We subjected these flies to EMS mutagenesis, followed by experimental evolution. After only 8–15 generations, experimental populations have normalized patterns of gene expression and increased survival. Using a phenomics approach, we find that populations were normalized through rapid increases in embryo size driven by maternal changes in metabolism and ovariole development. We extend our results to additional populations of flies, demonstrating predictability. Together, our results necessitate a broader view of regulatory network evolution at the systems level.

Changes in gene regulation underlie much of phenotypic evolution[1]. However, our understanding of regulatory evolution is likely biased[2], as most evidence is derived from observations of sparse natural variation or limited experimental perturbations[3], especially in a developmental context. Furthermore, developmental networks orchestrate multiple processes that span a range of organizational scales—from single cells to tissues and organs and to entire organisms[4]. These complex regulatory programs also integrate metabolic states[5] and environmental cues in response to complex ecologies[6,7]. However, developmental networks are often explored using a reductionist approach, focusing on particular time windows or pathways of development[8]. While such approaches have been foundational to our understanding of development, this narrow focus may have limited our understanding of other 'possible' paths of regulatory evolution that are not taken in nature[9]. Toward this goal, a 'synthetic evolution' method that employs mutagenesis and experimental evolution may be necessary to explore the evolutionary potential and constraints of developmental systems.

Quantitative genomics further challenges our models of how regulatory networks function—for complex traits, most of the heritability is likely due to a large number of variants, each with a small effect size[10]. Thousands of individual genes may contribute to phenotypes through expression in relevant cells[10], and the contributions of each genetic variant to developmental fates are often small and challenging to measure[11–13]. Therefore, it is essential to consider regulatory evolution and development both at the systems level and across populations[14–16]. Clearly, approaches to elicit the relationships between different phenotypic layers and how these changes manifest across populations are needed to understand the evolution of developmental regulatory networks.

In this study, we explored the well-characterized early embryonic segmentation network in *Drosophila*[17] in response to extra copies of *bicoid* (*bcd*), a key morphogen in *Drosophila* embryonic development. *bicoid* encodes a transcription factor whose mRNA is maternally deposited at the anterior pole of fly eggs. After fertilization, the Bicoid protein forms a concentration gradient across the anterior-posterior

[1]European Molecular Biology Laboratory (EMBL), Heidelberg, Germany. [2]Molecular Medicine Partnership Unit between EMBL and Heidelberg University, Heidelberg, Germany. [3]BioInnovation Institute, Copenhagen, Denmark. [4]Present address: College of Life Sciences, Beijing Normal University, Beijing, China. ✉e-mail: lixueying@bnu.edu.cn; justin.crocker@embl.de

(A-P) axis of the embryos, providing positional information for downstream targets, including gap and pair-rule genes such as *hunchback* (*hb*), *giant* (*gt*), *krüppel* (*Kr*), and even-skipped (*eve*). These genes and others together constitute a complex network that determines segmentation[18] and scaling[19,20] along the A-P axis of the embryo. It has been shown that the patterning outcomes of the network directly respond to the gene dosage of *bicoid*, with a higher dosage causing a posterior shift of the cephalic furrow and increased frequency of segmentation defects in larval cuticles[21]. We were able to directly monitor developmental changes that rescue or mitigate the phenotypic defects caused by altered gene expression and, in some cases, to even generate novel phenotypes. We found that compensatory changes for developmental perturbation can appear rapidly in the lab, with extensive phenotypic changes in gene expression, metabolism, and maternal anatomical features. Finally, we suggest that patterns observed in laboratory evolution can recapitulate phenotypic diversity in nature.

## Results

### Rapid population responses to extra copies of *bicoid*

The *bicoid* network in *Drosophila melanogaster* is one of the best-understood developmental networks[22]. Bicoid is a transcription factor that forms a concentration gradient along the anterior-posterior (A-P)

axis in the early embryo (Fig. 1a, b, Supplementary Movie 1). In embryos with two extra copies of *bicoid* (4x*bcd*, Fig. 1a–c), the cephalic furrow shifts toward the posterior[18], indicated by the expression of *eve*, an essential segmentation gene expressed in a striped pattern (Fig. 1d, e). Despite the positional defects, the 4x*bcd* embryos can develop into normal adults—albeit with an increased frequency of cuticle defects (Fig. 1f–h) and reduced viability to adulthood[21,23] (68.5%, Fig. 1i, Table S1).

The reduced viability of 4x*bcd* flies is a fitness disadvantage that can be a selection pressure in experimental evolution (Fig. 1j). To explore the system's capacity to respond to a perturbation of Bicoid levels, we established 15 parallel laboratory populations from 7 pools of chemically mutagenized 4x*bcd* flies (including replicates, see Fig. S1), along with three non-mutagenized populations which represent the standing variation in the lab stock. Based on whole-genome sequencing data, we estimated that the chemical mutagenesis with Ethyl methanesulfonate (EMS) introduced, on average, 2.7 point mutations per Mb. Thus, we estimated that the founding populations contained 1.7 million novel mutations (see "Methods", Fig. S1c), providing genetic diversity for selection. We set the 4th generation after mutagenesis as our starting point of experimental evolution, assuming that the generally deleterious mutations were purged in the first three generations. The mutagenized populations were maintained over

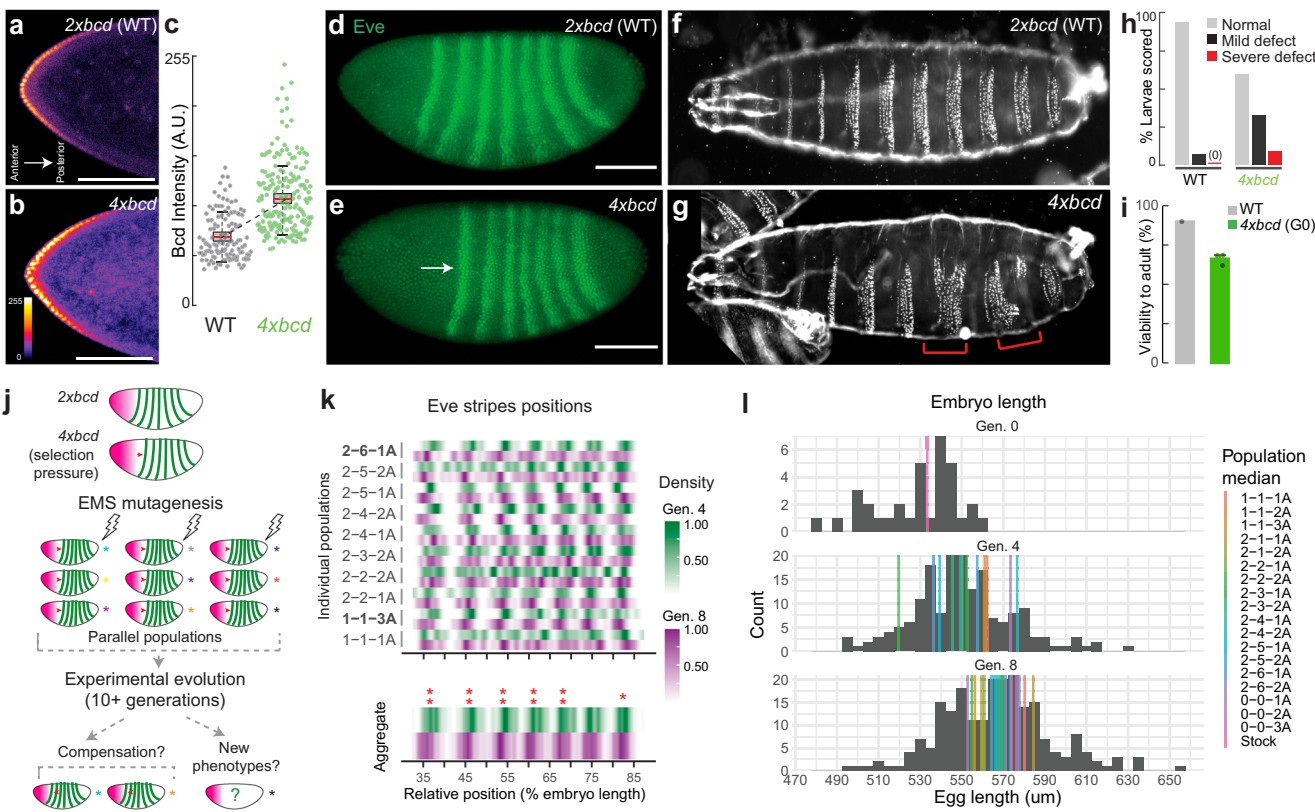

**Fig. 1 | Rapid changes of the *bicoid* network after experimental perturbation. a, b** Bicoid gradient in embryos with two (wild-type) or four copies of *bicoid* (anti-Bicoid immunostaining, stage 4 embryos). Scale bar = 100 um.). **c** Bicoid levels in the ten most anterior nuclei, quantified across 11 and 12 embryos for wild-type and 4x*bcd*, respectively. **d, e** Expression of *even-skipped* (*eve*) (anti-Eve immunostaining, stage 5 embryos), representative of 20–30 embryos. Scale bar = 100 um. **f, g, h** Cuticle phenotypes, with red brackets highlighting severe defects. Quantification in (**h**) was from 41 and 34 cuticles, respectively. **i** Viability to adulthood, presented as mean values +/- standard error. Viability was measured from -100 embryos per replicate, with 1–3 replicates per group. **j** Scheme of experimental evolution. **k** Distribution of *eve* stripes positions in mid-stage 5 embryos, detected by in situ hybridization. Top, individual populations (N = 3–22 embryos, with a median of 13).

Bottom, all populations aggregated (N = 60 for Generation 4, N = 217 for Generation 8). Intensity represents the scaled density of the designated population. Asterisks indicate significant shifts in the scaled position between generations. **, *p* < 0.01; *, *p* < 0.05 (two-sided Wilcoxon test, FDR-adjusted). **l** Distribution of embryo length across generations (gray histogram, all populations aggregated; N = 34, 176, and 217 for Generation 0, 4, and 8, respectively). Color bars represent the median of each population. Population 0-0-1 A, 0-0-2 A, and 0-0-3 A are non-mutagenized populations representing standing variation in the lab stock. G0 in (**i**) and (**l**) represents a non-mutagenized 4x*bcd* stock. All boxplots in this work are defined as follows: center line, median; box limits, the first and third quartiles; whiskers, 1.5x interquartile range. Source data are provided as a Source Data file.

generations to select for compensatory mutations that can rescue or mitigate the fitness defect. We primarily used *eve* stripe positions as an indicator for compensatory changes: the compensated embryos should show *eve* stripes positions shifted to the anterior of the ancestral 4x*bcd* line [37.2 ± 0.4% egg length (EL) for the first *eve* stripe, 95% confidence interval, Generation 4] and closer to the wild-type positions (28.3 ± 0.6% EL for the first stripe, VK33).

We found that compensation for the higher *bicoid* dosage occurred rapidly in our experimental populations. From the 4th to the 8th generation, the first *eve* stripe shifted to the anterior (toward the wild-type position) on average by 1.1% EL, from 37.2 ± 0.4% EL to 36.1 ± 0.2% EL (p < 0.01, Wilcoxon test) (all populations aggregated, Fig. 1k, **bottom panel**). Other stripes also showed different magnitudes of anterior shifts compared to Generation 4, ranging from 0.4% EL (stripe 7, p = 0.04, Wilcoxon test) to 1.0% EL (stripe 3 and 4, p < 0.01, Wilcoxon test) (Fig. 1k, **bottom panel**). Among these populations, there were heterogeneous responses in *eve* positions (Fig. 1k, **top panel**), with populations 1-1-3 A and 2-6-1 A showing significant compensatory shifts in more than one stripe in Generation 8 (Fig. 1k, Fig. S2a). Other populations showed different levels of shifts in *eve* stripes ranging from −2% EL to +2% EL (Fig. S2a), but the statistical power in detecting these shifts was low due to a limited sample size. We did not find a higher similarity between replicate populations from the same mutant pool than those from different pools. Interestingly, the compensatory shifts in population 1-1-3 A occurred through a shortened anterior region, whereas population 2-6-1 A compensated via an expansion in the posterior region, suggesting multiple possible mechanisms for compensation (Fig. S2b–f). These shifts could not be explained by the deactivation of *bicoid* copies because the *eve* positions in the evolved embryos were still much closer to those in the 4x*bcd* ancestors than in 2x*bcd*. Despite the seemingly subtle compensatory shifts, we note that a shift of 1% EL was the highest level of natural variation ever reported in *D. melanogaster*[24], suggesting that the early embryonic segmentation network can shift rapidly in the lab under directed selection. In addition, the experimental populations showed increased survival rates to eclosure after 16 generations (74.2 ± 2.5%, averaged across all populations) compared to the ancestral line (66.3 ± 3.4%), consistent with adaptation (Fig. S1d).

Unexpectedly, we found that compensation in the *bicoid* network coincided with an increase in egg length across the populations. From the 4th to the 8th generation, median embryo length increased from 550 um to 567 um (all populations aggregated, Fig. 1l, histogram, p = 1.81e-09, Wilcoxon test). Strikingly, despite variable embryo sizes, nine out of 12 populations showed an increase in median embryo length (1-1-1 A, 2-2-1 A, 2-2-2 A, 2-3-1 A, 2-3-2 A, 2-4-1 A, 2-5-1 A, 2-5-2 A, and 2-6-1 A; Fig. 1l, colored lines) and three of them (2-2-2 A, 2-5-1 A, 2-6-1 A) were statistically significant (p < 0.05, Wilcoxon test; Fig. S2c). This recurrent pattern suggests that an increase in embryo length might provide a quickly accessible mechanism to buffer the developmental stress caused by overexpression of *bicoid* and thus could drive the rapid compensatory changes we observed.

In parallel to phenotypic changes, we also found recurrent directional changes at the genomic level consistent with selection (Fig. S3). We performed low-coverage whole-genome sequencing for all 18 populations at the 3rd and 7th generation and focused on changes in allele frequency in common variants shared across populations (i.e. standing variation) to understand the population dynamics at a broad scale. We found 16,394 biallelic variants showing consistent increases or decreases in allele frequency in two or more populations (Fisher's exact test, FDR-adjusted p < 0.05, Supplementary Data 1). Based on a sign test, 181 of them were biased toward being maintained or purged in six or more populations (Fig. S3c). Recurrent gain or loss of these alleles across multiple populations could suggest selection. For example, a non-synonymous mutation in Melted (F21V) was purged in six populations at the 7th generation (Fig. S3d), which could be

beneficial because *melted* was linked to growth and metabolic pathways, and its mutant showed nutrient deprivation[25]. Other variants potentially under directed selection include those related to metabolism (e.g. *Apoltp*, Supplementary Data 1) and ovariole development (*e.g. mtgo, bru3*, Fig. S3d, Supplementary Data 1)[26]. These changes in allele frequency are consistent with rapid adaptation in the laboratory populations, with possible links to maternal and metabolic-related genes.

## Compensation through an increase in embryo length

To further address the possible link between embryo size and the *bicoid* network, we focused on population 2-6-1 A to dissect the developmental changes before and after laboratory evolution. In this line, *eve* stripes consistently shifted to the anterior in the 8th and the 10th generation compared to the 4th generation (Fig. 2a–c; Fig. S4a), with the shift of the last stripe being the most prominent (Fig. 2c, d). We found that the shifts occurred simultaneously with an expansion of the posterior region: the egg length was consistently longer in both generations (540.5 ± 6.5 um at Generation 4, 573.5 ± 13.6 um at Generation 8, and 560.4 ± 7.1 um at Generation 10; Fig. 2e, Fig. S4b). The expression of *tailless*, a gap gene that specifies the posterior identity, was also wider in the 8th generation than in the 4th generation (Fig. 2f–i). While the total number of nuclei along the A-P axis has not significantly changed (Fig. S4c), consistent with early embryos' limited capacity to regulate cell number[27], there was a slight increase in the number of nuclei in the posterior region, from *eve* stripe 7 to the posterior pole at Generation 8 (12.3 ± 0.9 vs. 14.1 ± 1.1, p = 0.048, Wilcoxon test, Fig. 2j), as well as an overall increase in the distance between nuclei (6.39 ± 0.23 um vs. 6.82 ± 0.13 um, p = 0.004, Wilcoxon test, Fig. S4d, e). Consistent with compensatory changes, the line has stabilized phenotypes across phenotypic scales, including cuticle phenotypes (Fig. 2k) and viability to adulthood after 15-16 generations (Fig. 2l).

The compensation via embryo size appeared to be relatively short-term, because the embryo length of population 2-6-1 A peaked at Generation 8 and 10, but gradually reduced after Generation 15 and resumed wild-type level at Generation 49 (Fig. S4a, b). This could be due to the fact that overly large embryos might have deleterious effects and cannot persist as a long-term solution in the standard environmental conditions employed in this work. Such a turnover in adaptive strategies is not uncommon in evolution[28–31]. Future research along these lines could reveal alternative strategies to compensate for high *bicoid* dosage that is independent of embryo size, such as the response of Population 1-1-3 A, which showed a shortened anterior region (Fig. 1k, Fig. S2).

Together, these data lead us to hypothesize that the compression of the trunk and tail caused by extra Bicoid might be mitigated in larger embryos due to more space in the posterior region. These results are consistent with previous findings on the interaction between egg size and the *bicoid* network[24,32,33]. Furthermore, because egg size is a highly polygenic and evolvable trait[34–36], it might have provided a large capacity to respond rapidly to genetic and environmental changes.

## Changes in maternal metabolism and ovariole development

To identify possible molecular bases that can support the rapid phenotypic stabilization through changes in egg length, we performed single-nuclei transcriptomics with early embryos in the evolved line (2-6-1 A, Generation 20) (Fig. S5, Table S2). The evolved line had a striking increase in the proportion of yolk nuclei compared to wild-type or the 4x*bcd* lab stock (6% vs. 1%, p < 0.001, fisher's exact test, Fig. 3a), consistent with the increased nutritional need of larger embryos. Among marker genes of the yolk cluster, there were 230 genes differentially expressed in the evolved line, including those related to metabolism (*bmm, trbl, Lime, Srr*) and cell growth (*crp, Traf4*) (Fig. 3b, c, and Supplementary Data 2). Previous research suggests that the *Drosophila* body/organ size can be directly controlled by signaling pathways

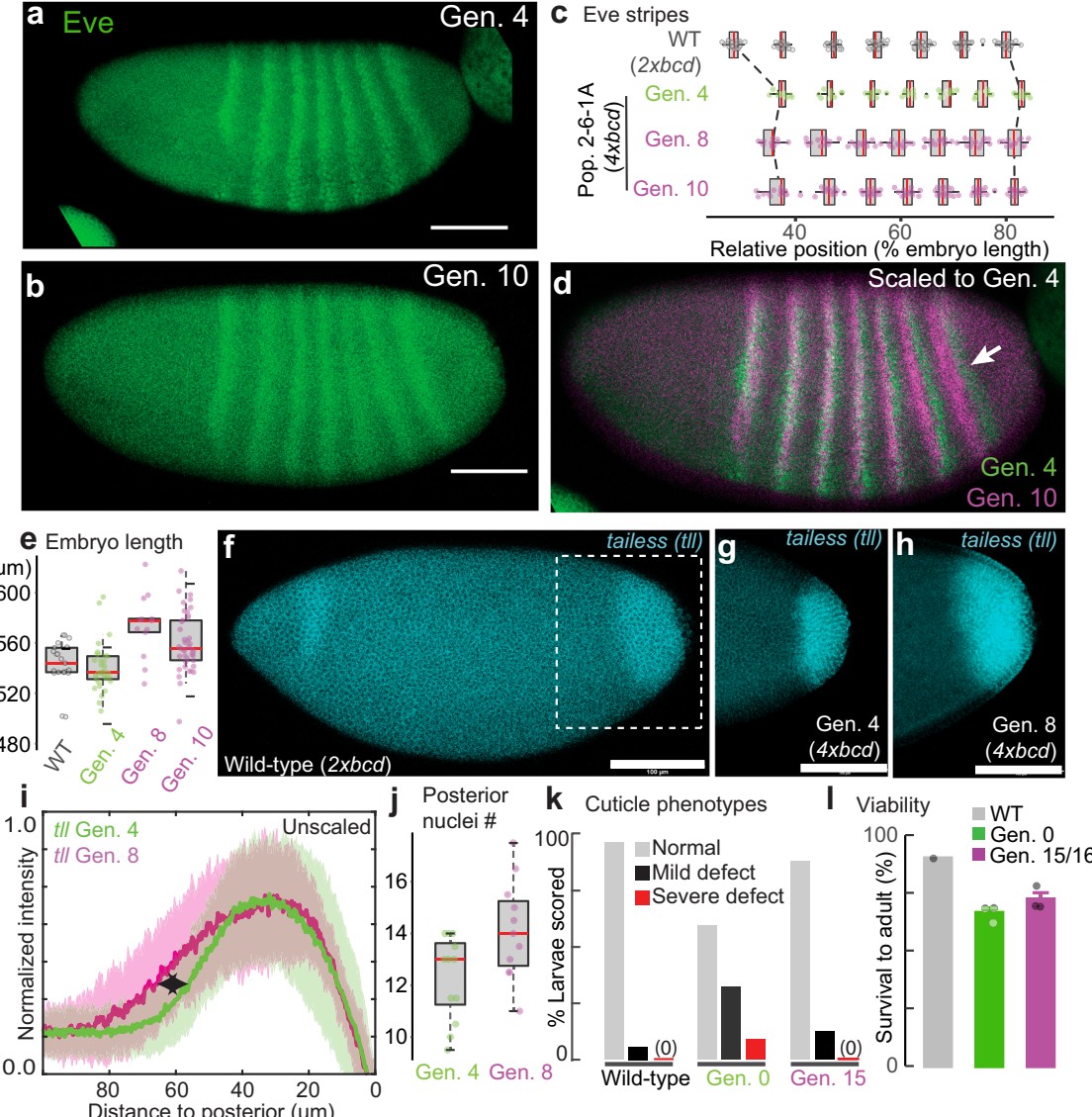

**Fig. 2 | Compensatory changes in gene expression, embryo length, cuticle, and viability. a–d** Eve stripes in Population 2-6-1 A (anti-Eve staining), with the arrow in (**d**) showing a prominent anterior shift in the 7th stripe. The shifts were quantified in (**c**) from in situ data (*eve* co-stained with *sna*). N = 21, 12, 13, and 16 for WT, Gen. 4, Gen. 8 and Gen. 10 respectively. **e** Embryo length. N = 21, 33, 13, and 41 for WT, Gen. 4, Gen. 8 and Gen. 10 respectively. **f–i** *tailless* (*tll*) expression, detected by in situ hybridization. **i** shows the normalized intensity profiles aligned at the posterior end. Solid lines are average *tll* intensity and the shaded panels denote the standard deviation. N = 22 and 14 for Gen. 4 and Gen. 8, respectively. **j** The number of nuclei from the posterior boundary of *eve* stripe 7 to the posterior pole. N = 12 and 11 for Gen. 4 and Gen. 8, respectively. **k** Rescue of cuticle defects. N = 41, 34, and 32 larvae. **l** Viability to adulthood, presented as mean values +/- standard error. Viability was measured from ~100 embryos per replicate, with 1–3 replicates per group (also see Fig. S1). Scale bar = 100 um. G0 in (**k**) and (**l**) represents a non-mutagenized *4xbcd* stock. All boxplots in this work are defined as follows: center line, median; box limits, the first and third quartiles; whiskers, 1.5x interquartile range. Source data are provided as a Source Data file.

involved in metabolic regulation and cell growth, such as the insulin signaling pathway[37,38]. We found a number of metabolic genes differentially expressed in the evolved line across multiple cell types, including epidermal ('*ovo*'), trunk ('*opa*'), anterior ('*oc*'), and posterior ('*byn*') clusters in the ectoderm, as well as in mesoderm, endoderm, yolk and pole cells (Fig. 3d, Supplementary Data 3).

The changes in yolk content and gene expression might imply a broader change in maternal metabolism to direct more nutrients into the eggs, and thus enable larger embryo sizes. Indeed, we found that the evolved embryos contained more triglycerides (TG) than two wild-type lines (Fig. 3e). Triglycerides are essential components of yolk-related lipid droplets[39] that can act as metabolic fuel for *Drosophila* embryogenesis[40], and high triglyceride levels have been linked to bigger embryo size in multiple animals[41,42]. To further characterize this

metabolic alteration, we performed MALDI-imaging mass spectrometry (MALDI-IMS) in positive ion mode[43] on cryo-sectioned slices of ovaries. This technique allowed us to reconstruct entire mass spectra for single oocytes, and thus trace this phenomenon back to the oocyte stage. We found differences in the lipid signature of oocytes between the evolved *4xbcd* line (2-6-1 A, Generation 42) and wild-type (w1118) (Fig. S6a), including elevated levels of triglycerides and decreased levels of glycerophosphocholines in the evolved line (Fig. 3f, g, Fig. S6b, c). Additionally, there were global differences in the fatty (FA) distribution in the evolved line, showing a higher abundance of FAs with 13, 14, and 15 carbons, and reduced levels of FAs with 18 carbons on their chain (Fig. 3g). This observation was confirmed by tandem mass spectrometry coupled with MALDI-IMS in negative ion mode, which independently detects a wide range of lipid ions

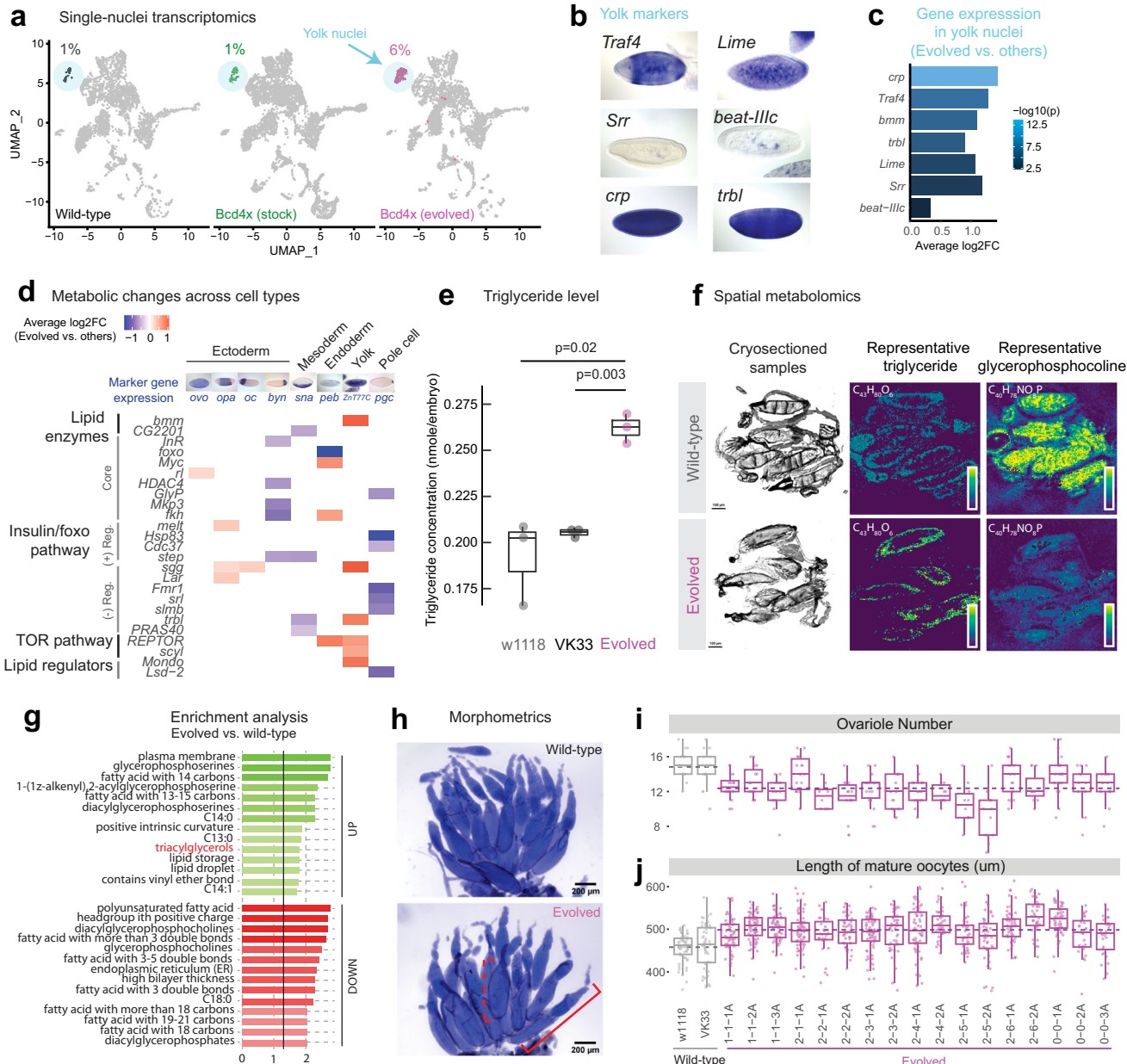

**Fig. 3 | Phenotypic changes in gene expression, metabolism and ovariole development in the evolved lines. a** UMAP of single-nuclei transcriptomes of stage 5 embryos (see Fig. S6 for details). The colored clusters show yolk nuclei. Wild-type is VK33. The evolved line is population 2-6-1 A at Gen. 20. **b** Representative marker genes of yolk nuclei. **c** Representative marker genes of yolk nuclei that were differentially expressed in the evolved line. *P*-values were based on a two-sided Wilcoxon Rank Sum test followed by Bonferroni correction. **d** Changes in expression of metabolic genes across cell types between the evolved line and the other two samples. Only significant changes (adjusted p-value < 0.05) are shown. ( + ) Reg., positive regulators; (-) Reg., negative regulators. FC, fold change. Images of marker gene expression in (**b**) and (**d**) are from BDGP in situ database[80]. **e** Enzymatic determination of triglyceride levels in stage 5 embryos (Gen. 50 for population 2-6-1 A). Points represent values from independent homogenates made from 50 embryos each. P values are from a two-sided Student's t-test. **f** MALDI-IMS of wild-type (w1118) and evolved (2-6-1 A from Gen. 42) ovaries. Left, middle sections from ovaries employed in MALDI-IMS. Scale bar = 100 μm. Middle, spatial distribution of a representative triglyceride, TG(40:1) at m/

z = 715.5846. Right, spatial distribution of a representative glycerophosphocholine, PC(32:1) at m/z = 732.5537. Abundance was normalized to TG(44:3) at m/z = 767.6159 which showed constant levels across all experiments. **g** Enrichment analysis based on the abundance values for 122 lipids detected through MALDI-IMS (same experiments as **f**). N = 9 oocytes for wild-type, N = 10 for the evolved line. The vertical solid line indicates a cutoff at FDR q-value of 0.05. Triacylglycerols were highlighted in red. **h** Ovaries of wild-type (w1118) and evolved (2-6-1 A, Gen. 39) lines, stained with DAPI. The solid red bracket indicates an ovariole, and the dashed red bracket indicates a mature oocyte. Scale bar = 200 um. **i** Ovariole number and (**j**) length of mature oocytes of wild-type and the evolved lines (Gen. 39). The horizontal dashed lines represent the mean of all wild-type/evolved lines aggregated (*p* = 9.783e-09 for ovariole number and *p* < 2.2e-16 for oocyte length, two-sided Wilcoxon test). N = 8–20 ovaries for ovariole counts. N = 27-72 oocytes for oocyte length. All boxplots in this work are defined as follows: center line, median; box limits, the first and third quartiles; whiskers, 1.5x interquartile range. Source data are provided as a Source Data file.

(Fig. S6d–f; also see "Methods"). Overall, these results show that the line has altered its lipid metabolism in a way that is consistent with bigger embryo sizes and higher energy requirements.

The changes in gene expression and lipid composition suggest rapid physiological changes at the maternal level. We examined the ovaries of the experimental populations and found that they tended to have fewer ovarioles ($12.4 \pm 0.3$ vs. $14.8 \pm 0.7$, all populations aggregated vs. wild-type aggregated, same below) and longer oocytes ($498.4 \pm 2.9$ um vs. $458.0 \pm 8.2$ um) than wild-type lines (Fig. 3h–j), consistent with a previous report that the egg size difference between *Drosophila* lines originated from oogenesis[44]. Therefore, the compensation could occur through a trade-off between ovariole number and oocyte size[45], possibly through growth-related mechanisms such as the insulin pathway[34,46]. Furthermore, we found that the change in size was specific to oogenesis and likely to have metabolic rather than behavioral underpinnings because we did not observe significant differences in larval length or larval feeding behavior (Fig. S7)[36].

## Predicting phenotypes from additional fly lines

Embryo size is known to vary widely within and between *Drosophila* species[24] and across environments[35]. As such, changes in embryo size could provide a way to rapidly mitigate the effects of Bicoid dose. To test if our observations could be extended, we examined two inbred lines isolated from the wild, Ind and Canton-S, with the former having larger embryos than the latter[24] (Fig. 4a). The anterior Bicoid concentration was higher in the larger Ind embryos (Fig. 4b, c), consistent with previous results[44,47–49]. These two natural isolates also show differences in ovariole number and oocyte length (Fig. 4d), as well as the level of triglycerides (Fig. 4e). Collectively, these observations suggest that the coupling among the *bicoid* network, egg size, maternal physiology, and metabolism could also exist in nature.

Next, to test if the bigger embryo size of the Ind genetic background could relieve the stress on the developmental network elicited by Bicoid overexpression, we crossed the *bicoid* transgenes into these inbred lines. In the crosses, the F1 offspring

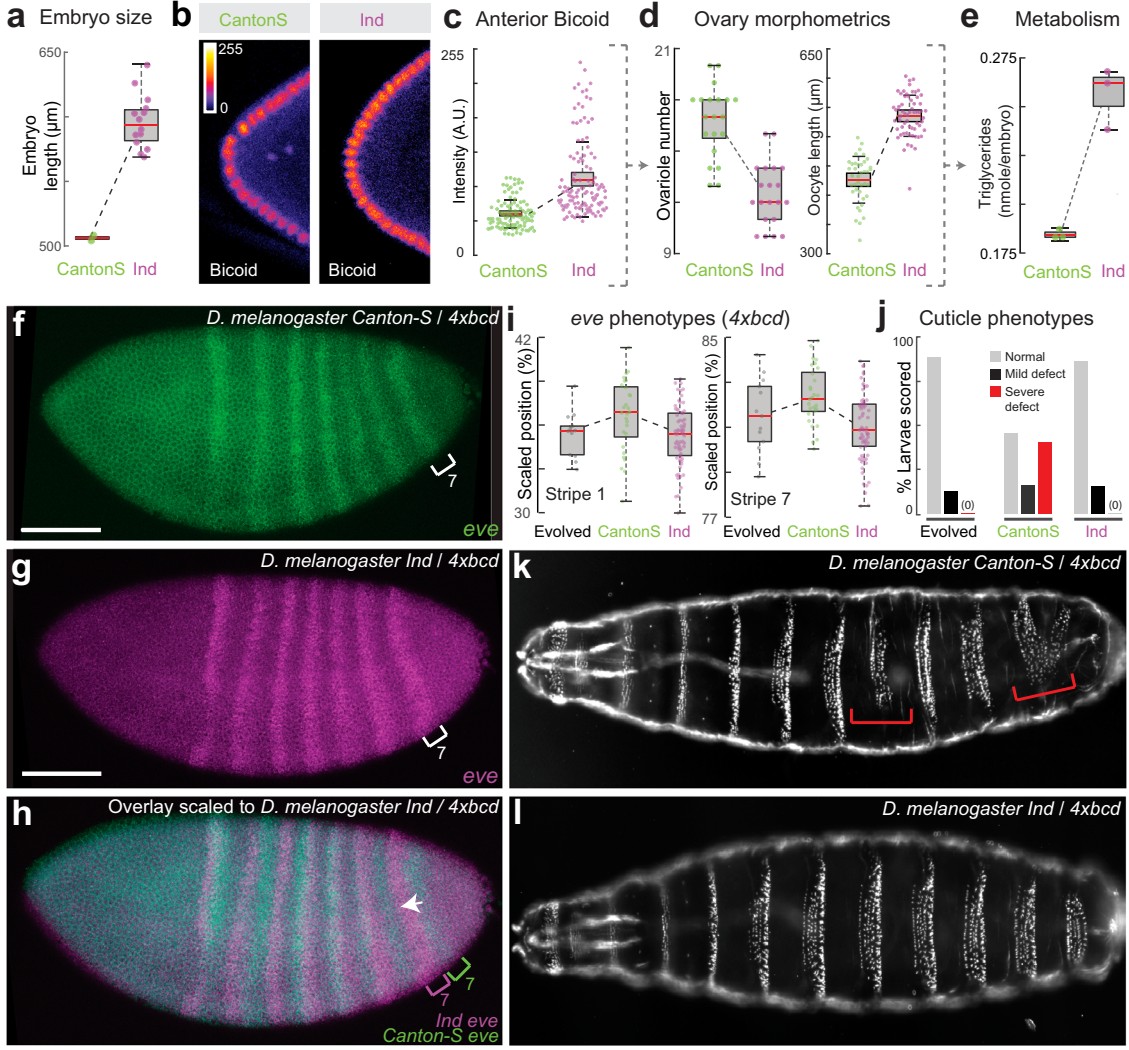

**Fig. 4 | Wild populations' responses to extra copies of *bicoid* and model for adaptation. a** Embryo size and (**b**, **c**) anterior Bicoid concentration (anti-Bicoid staining) of Ind and Canton-S. Each point in (**c**) represents one nucleus, quantified across 18 and 10 embryos for Ind and Canton-S, respectively. **d** Ovariole number (N = 20 ovaries for both groups), oocyte length (N = 39 and 62 oocytes for Canton-S and Ind respectively), and (**e**) level of triglycerides per embryo of Ind and Canton-S. Points in (**e**) represent values from independent homogenates made from 50 embryos each. (**f**–**i**) *eve* stripe positions of Ind (N = 72) and Canton-S (N = 28) when carrying 4xbcd. Scale bar = 100 um. The evolved line in (**i**) is 2-6-1 A at Generation 8 (N = 13, same data as Fig. 2c). (**j**–**l**) Cuticle phenotypes. The red brackets highlight severe defects. Quantification in (**j**) was from 32–48 cuticles. See Fig. S8 for full data. All boxplots in this work are defined as follows: center line, median; box limits, the first and third quartiles; whiskers, 1.5x interquartile range. Source data are provided as a Source Data file.

have 50% of genetic information from the wild isolates and have two extra copies of *bicoid* inserted on the second and the third chromosomes, respectively (4x*bcd* in total, see Fig. S8a for the crossing scheme). We also crossed them to a wild-type lab strain (VK33) to control for background effects. We found that embryos from F1 individuals in Ind/lab background were larger than those in Canton-S/lab background (Fig. S8b), suggesting that the Ind background had a dominant effect on embryo size. The *eve* stripes in Ind/lab background were located further to the anterior than the Canton-S/lab background in the control crosses (2x*bcd*) (Fig. S8c), suggesting natural variation in the capacity for scaling of the network. Such variation might be in favor of buffering stresses such as overexpression of *bicoid* - the difference was also present in embryos with 4x*bcd*, with the *eve* stripes of Ind embryos being anterior to those of Canton-S embryos, i.e. closer to the wild-type positions (Fig. 4f–i). Interestingly, the positions of *eve* stripes (Fig. 4i) and cuticle phenotypes (Fig. 4j–l) of 4x*bcd*-Ind embryos resembled those of population 2-6-1 A. 4x*bcd* embryos in the Ind background also had higher viability to adulthood compared with those in Canton-S or lab background (Fig. S8d), consistent with a higher tolerance of *bicoid* over-expression in larger embryos. Together, the evolved line is similar to Ind across a number of key phenotypes, supporting the hypothesis that changes in maternal contributions to embryo sizes could be used to buffer the dosage of *bicoid*.

The trends we found from experimental evolution, genetic perturbations, and the findings from the larger *D. melanogaster Ind* line, are all in line with evidence that *Drosophila* can adapt rapidly to laboratory culture on ecological timescales[50]. To explore the broader context of these results, we looked across a number of closely related *Drosophila* species (Fig. 5a, Fig. S9), testing the relationship between ovariole number and oocyte lengths (Fig. 5b). Consistent with previous results[36,45], we see a strong correlation across the *Sophophora* subgenus

indicating that such a trait may be consistent across a broader evolutionary context.

## Discussion

Little is known about how organisms respond to developmental perturbations in short timescales. The early segmentation network downstream of Bicoid has been characterized as a highly dynamic[51] yet robust network to ensure precise scaling of gap gene boundaries[20,24,47,52]. Perturbations to the network, such as a change in *bicoid* dosage, can lead to substantial patterning defects and fitness disadvantages[21] (Fig. 1). Leveraging the fitness disadvantage as a selection pressure provided us an opportunity to examine the robustness and evolvability of developmental systems. We found compensatory phenotypic changes within 8-15 generations, reflected in gene expression, larval morphologies, and survival to adulthood (Figs. 1–2). These results are consistent with the recent findings that adaptation in *Drosophila* was evident over only one to four generations in response to environmental changes, including changes in egg-size[50]. Such rapid phenotypic adaptation and large allele-frequency shifts over many independent loci in response to developmental changes may be a common mechanism for gene-regulatory network evolution[50].

One prominent phenotypic response to extra copies of *bicoid* was a general increase in egg length (Fig. 1l & 2e). What could be the mechanism for the large embryos to ameliorate the fitness defect caused by 4x*bcd*? One possibility is that the enlarged posterior region relieved some of the compression in the abdominal region, which has been associated with a lack of cell death[21]. The observed change in the posterior was also consistent with a recent study showing that posterior boundaries in *Drosophila* embryos are highly dynamic and sensitive to gene dosage[53]. Furthermore, the compensation could occur through an altered distribution of important factors in the segmentation network, which consists of maternal factors (*bcd*, *nos*, *tor*), gap

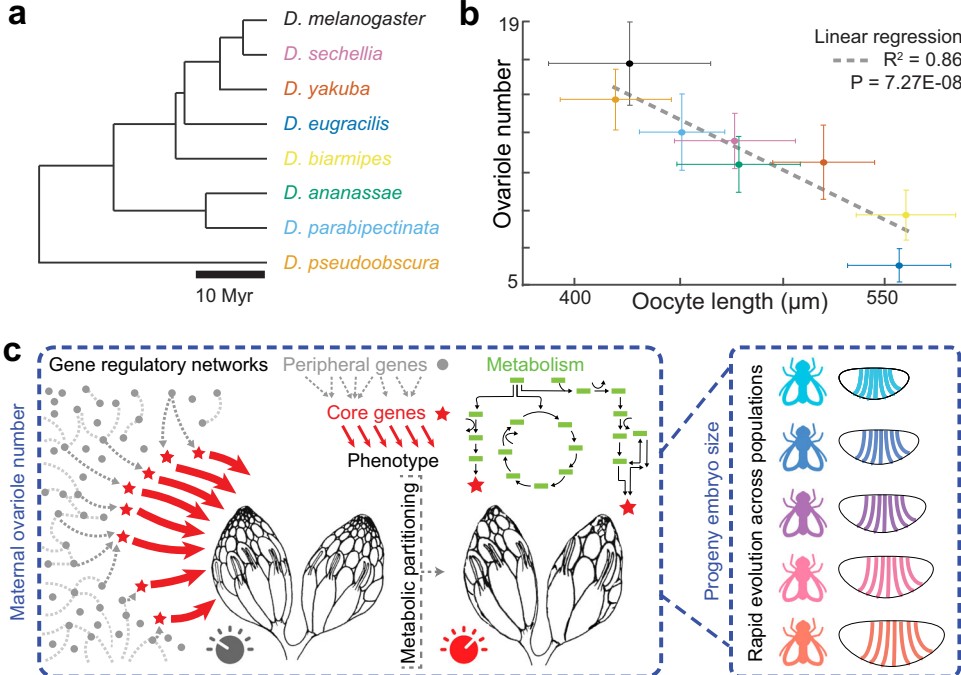

**Fig. 5 | Laboratory evolution predicts phenotypes of wild species. a** Phylogeny of species tested[88]. **b** The relationship between ovariole number (N = 14-20 ovaries) and oocyte length (N = 17-133 oocytes, with a median of 52.5), presented as mean values +/- standard deviation; colors are indicated in (**a**). *D. melanogaster* was represented by Canton-S. **c** Model for maternal compensatory changes in laboratory evolution. The embryonic patterning network is connected to a broad gene

regulatory network via core genes[89] (red stars) involved in maternal metabolism that tunes the size of ovarioles. Part of (**c**) is reprinted from Cell, Vol 177, Xuanyao Liu, Yang I. Li, Jonathan K. Pritchard, *Trans* Effects on Gene Expression Can Drive Omnigenic Inheritance, pages 1022-1034, Copyright (2019), with permission from Elsevier. Source data are provided as a Source Data file.

genes (*hb, gt, kni, Kr*) and pair-rule genes (*eve*, etc). Previous studies have shown that, when *bcd* dosage changes over a fivefold range, the expression boundaries of downstream gap genes and pair-rule genes change over a much smaller magnitude (up to 2-fold)[54], highlighting the importance of additional maternal factors such as Nanos or Torso, as well as cross-regulation among the gap genes themselves. Therefore, it would be of future interest to quantitatively characterize the dynamics of these factors over generations to understand how the evolutionary compensation occurred.

Additionally, understanding the compensatory mechanism for 4x*bcd* would be promising in revealing unknown properties of the segmentation network. In wild-type embryos, the gap genes and pair-rule genes scale at 1% precision within a short developmental time window, using an unscaled Bcd gradient as input[55,56]. Theoretical models suggest that the information needed for the precise scaling is sufficiently encoded in the above-mentioned factors[57]. In this study, we found a shift of 1.1% EL in *eve* stripe positions under 4x*bcd* selection, which seemingly "escaped" the precise control of the network. This finding is in line with previous observations that embryonic geometry can affect the scaling of gap gene boundaries under perturbations, including genetic manipulation[32] and artificial selection[33,47,49]. Given further molecular characterization, these perturbed systems have the potential to provide new models for network dynamics, incorporating the interactions between the embryonic size-control network and the segmentation network[44,48,49].

The rapid phenotypic compensation driven by embryo size is likely related to its genetic architecture. Egg size is a trait known to be both highly polygenic[34] and evolvable in both common garden experiments[33,50] as well as across natural populations[24,35,36]. As such, the egg-size network might provide a much larger set of targets for selection than targets directly downstream of Bicoid, and hence the change in egg length appeared as the first response in a short evolutionary timescale. These results are consistent with models that posit that phenotypic evolution may be driven by many loci of small effect[58,59]. Furthermore, the rapid changes were associated with changes in ovariole number, which is also known to be controlled by many genes[26], resulting in changes in metabolism and embryo size. Therefore, there could be numerous genes at different phenotypic levels that provide evolutionary accessibility to compensation. It is possible that the segmentation network, which can readily scale within and between species[20], is the result of selection for a highly evolvable system that provides developmental plasticity for early embryos across variable ecologies[60] (Fig. 5c).

Our study is subject to a few limitations, highlighting the challenges in longitudinal studies of laboratory populations. In our experimental-evolution design, we set up multiple parallel-evolving populations with an intensive sampling schedule with the aim to characterize network dynamics at scale. However, despite our efforts in high-throughput embryo-handling and automated imaging[2,61], we were often limited by technical factors such as sample size, batch effects, and drift. A higher level of automation would allow systematic examinations of different adaptive strategies in parallel populations (e.g., compensatory mechanisms other than embryo size) and exploration of the generalizability of the proposed model. One of the challenges is that random mutagenesis introduces many mutations that may be both unrelated and highly deleterious[62]. Further, mapping causal variants, which can be broadly distributed with low-effect sizes, remains a challenge[63]. Therefore, in the future, more targeted in vivo mutagenesis approaches biased towards gene regulatory networks can be developed for the study of the genetics and evolution of the *Drosophila* regulatory genome.

The phenotypic differences were not limited to early embryonic development but included changes in lipid metabolism (not only increased yolk content and triglyceride levels, but also changes in the relative abundance of physiologically relevant phospholipids), cell-type-specific gene expression (rewiring of metabolic gene network), and maternal anatomy (reduced ovariole numbers) (Fig. 3). These results show that perturbation of one node of the developmental network, the *bicoid* dosage, can lead to profound organism-wide responses across multiple phenotypic scales. Importantly, these observations highlight the deep connections between multiple phenotypic layers of multicellular systems and argue for a broader 'phenomics' perspective[16], instead of a strictly gene-centric view. Exploring the interplay of metabolic and developmental networks could transform our understanding of evolution and development across variable ecologies[5,15], as such processes are fundamentally linked[64]. In the future, synthetic evolution approaches using animal systems could provide a generalizable platform for the dissection of gene regulation and complex genomes.

## Methods
### Fly genetics
The eGFP-Bicoid fusion construct was designed according to Gregor et al.[65] (see Supplementary Data 4 for the construct map). The construct was synthesized and cloned into placZattB by Genscript, and was transformed into *D. melanogaster* at the VK18 or VK33 landing site following standard PhiC31 integrase protocol, with the help of injection service provided by Alessandra Reversi at EMBL. The transformants at the VK33 site were homozygosed by sibling crosses to construct a stable 4x*bcd* line and subsequently used in mutagenesis and experimental evolution.

We also established balancer stocks from the transformants at VK18 (second chromosome) and VK33 (third chromosome) sites, and used them to generate a 6x*bcd* line, with an extra copy of *bicoid* on each of the second and the third chromosomes.

To examine the response to extra copies of *bicoid* in wild populations, virgins of Ind ("Mysore" strain, old stock #3114.4 from National Drosophila Species Stock Center, US) and Canton S (Bloomington stock #64349) were crossed to 6x*bcd* males. The F1 flies are heterozygous for the alleles from the wild populations and carry two extra copies of *bicoid*. They were used to set up egg-collection chambers and the F2 embryos were examined for *eve* expression, cuticle phenotypes, and fitness (Fig. S8a). To control for background effects, the natural isolates were crossed to the VK33 stock, which has the same background as the 6x*bcd* line.

At Generation 40, we outcrossed 2-6-1 A males to wild-type w1118 or VK33 for four generations. In each generation, males with orange eyes (heterozygous for the *egfp-bicoid* transgene) were crossed to virgins of w1118 or VK33. After four generations, males and virgins with orange eyes were mated, and their progeny were selected for homozygotes (red eyes) to create 'new' 4x*bcd* lines. In this way, we expect to remove or 'dilute' 2-6-1A-associated mutations and study the effects of 4x*bcd* without any compensatory evolution.

The non-*melanogaster* species were a generous gift from Nicolas Gompel, with the exceptions of *Drosophila parabipectinata* which was kindly provided by Artyom Kopp, and *Drosophila virilis*, which was kindly provided by Eileen Furlong. Strain background: *D. ananassae* (TSC 14024-0371.13), *D. biarmipes* (TSC 14023-0361.01), *D. eugracilis* (from the US National *Drosophila* Species Stock Center), *D. parabipectinata* (inbred derivative of strain TSC 14024-0401.02), *D. pseudoobscura* (TSC 14011-0121-94 USA), *D. sechellia* (TSC 14021-0248-25), *D. yakuba* (TSC 14021-0261.01) and *D. virilis* (*w*).

### Mutagenesis and experimental evolution
EMS-mutagenesis was performed according to Bökel (2008)[66]. Briefly, around 1,000 4x*bcd* male flies (G0) were fed with 1% sucrose solution containing 25 mM EMS, and were then mated to 4x*bcd* virgins. Around 3,500 F1 flies were used to establish 7 independent mutant pools, with 400-600 flies per pool. Specifically, the mutagenesis was done in two batches: flies from the first batch were used to establish one mutant

pool, labeled 1-1, and flies from the second batch were used to establish six mutant pools, labeled 2–1 to 2–6. The mutation rate did not obviously differ between the two batches based on subsequent genomic analysis (see below).

Each mutant pool was used to seed 2-3 bottles of progenies consecutively ('set A') and these bottles were replicated at the 3$^{rd}$ generation ('set B'), to provide 4-6 replicate populations in total for each mutant pool (Fig. S1a). For example, Pool 1-1 was used as parents to produce Populations 1-1-1 A, 1-1-2 A and 1-1-3 A, by transferring the parents to a new bottle every 4-5 days. F3 flies from these populations were used as parents to produce Populations 1-1-1B, 1-1-2B, and 1-1-3B, respectively. Populations in set B were primarily for backups in this study.

The flies were maintained at 25 °C under standard fly-rearing condition under non-overlapping generations, to select for rescuing mutations (standard fly food recipe in our facility: 180 g yeast, 100 g soy flour, 800 g cornmeal, 800 g malt extract, 120 g agar, 160 ml sugarbeet sirup, 62.5 ml propionic acid, 120 ml 20% nipagin ethanol solution, and 10 ml water). The population size was approximately 200-500 for each generation. Three populations of non-mutagenized 4xbcd flies were maintained under the same condition for comparison (labeled 0-0-1 A, 0-0-2 A and 0-0-3 A). During the first 15 generations, the populations were sampled every 2-5 generations for embryo collection, and the adult flies were frozen for genomic DNA (Fig. S1b).

### Embryo fixation, antibody staining and fluorescent in situ hybridization

*Drosophila* embryos were fixed and stained following standard protocols[67]. In particular, stage-5 embryos were acquired from a 5-hr egg-laying window at room temperature. A fixation time of 18 min was used for these embryos to adapt to the sensitivity of Eve antibody. The Eve antibody (mouse, Developmental Studies Hybridoma Bank, 2B8-concentrate) was used at 1:20 dilution. Bicoid antibody (rabbit) was a gift from Pinar Onal and Stephen Small, and was used at 1:250. DIG-, FITC- or biotin-labeled, antisense RNA-probes were used to detect gene expression of *eve*, *sna*, or *tll*, respectively (see Table S3 for primer sequences). All embryos were mounted in ProLong Gold with DAPI, and imaged on a Zeiss LSM 880 confocal microscope under 20x (air, 0.8 NA) or 25x (oil, 0.8 NA) objective.

### Image analysis

All images were rotated to orient along the A-P axis before analysis, with the A-P axis positioned horizontally and the dorsal-ventral (D-V) axis positioned vertically (see Fig. S2d for an example). All image analysis in this study was performed with Fiji (ImageJ 1.54 f). All data analysis regarding organismal phenotypes was performed in R (4.3.2).

**Position of eve stripes.** Images from fluorescent in situ hybridization of *eve*, *snail* (*sna*) and *tailless* (*tll*) were used to quantify *eve* position precisely. Embryos were imaged as Z-stacks, with the measurements performed on the Z-slice where *eve* and *sna* were in focus. We manually extracted the positions of the intersection of *sna* expression and the anterior boundary of each *eve* stripe in mid-stage 5 embryos (see Fig. S2d for an example), staged based on the degree of membrane invagination. The use of *sna* to mark a particular dorsal-ventral position on the *eve* stripes enabled precise quantification of the *eve* positions, which could also explain the differences between our results on Ind and Canton-S and a previous publication[24].

**Embryo length.** Embryo length was manually extracted from Z-stacked confocal images, from anterior to posterior, excluding the pole cells.

**Bicoid concentration.** Bicoid intensities were acquired from anti-Bicoid staining by extracting the average nuclear intensity for ten nuclei at the anterior pole for each embryo, as per Dubuis et al. [68].

**Tll profiles.** The intensity profiles were extracted from a rectangular region of 3-4 cells' height along the *A-P* axis from max-projected confocal images[69], normalized to peak intensities. The dorsal-ventral position was determined using the border of *sna* expression.

**Nuclei counts.** The number of nuclei along the A-P axis was counted along the *sna* border independently by two experimenters (X.C.L and L.G.), on one Z-slice where *eve* and *sna* were in focus. In the posterior region where *sna* is not expressed, we counted the nuclei along the extension line of the *sna* border all the way until the posterior end (excluding the pole cells). The counts from the two experimenters were not significantly different. Numbers from the two experimenters were averaged for each embryo. Particularly, the total number of nuclei (left panel in Fig. S4c) were averaged across two measurements by X.C.L. and one measurement by L.G, whereas the nuclei in the most anterior and most posterior regions (middle and right panels in Fig. S4c) were only counted once by each experimenter.

**Nuclei distance.** While counting the nuclei (see above), we marked the center of each nucleus and extracted their x-y coordinates in ImageJ, in order to calculate the 2D-distance between neighboring nuclei along the A-P axis: $D = \sqrt{(x_1 - x_2)^2 + (y_1 - y_2)^2}$, where $x_1$ and $x_2$ represent the x coordinates of two neighboring nuclei and $y_1$ and $y_2$ represent the y coordinates of them. The average inter-nucleus distance was calculated using all nuclei counted along the A-P axis and across two experimenters' measurements for each embryo (Fig. S4d). Additionally, we plotted the inter-nucleus distance ($D$) as a function of the nucleus position ($x_2$) along the A-P axis (Fig. S4e), which showed that the difference between F4 and F8 embryos was mainly in the anterior and middle regions of the embryos.

### Cuticle preparation

Overnight embryos were collected, bleached, rinsed and transferred into clean water in a petri dish, where they were allowed to develop for 24 h at room temperature. After 24 h, the larvae were transferred onto a glass slide and mounted in Hoyer's medium mixed with lactic acid (1:1). The slides were baked in an oven at 55 °C for 48 h and were then imaged with dark field microscopy using a Zeiss M2 compound microscope.

The cuticle images were scored based on the criteria from Namba et al. [21]: severe defect – fusion or missing segments; mild defect – missing or misaligned denticles in any segment; normal – no visible defects. w1118 was used as wild-type.

### Survival assay

Around 100 embryos from an overnight plate were manually transferred onto an apple juice plate with yeast in the center and left at room temperature for 24 h. On the second day, the number of unhatched embryos was counted for each plate, and the entire agar (with larvae and unhatched embryos) was transferred to a food vial. The eclosed adults were counted from day 12 until no adults came out. All the survival assays were performed at room temperature.

### Whole-genome sequencing

**Genomic DNA extraction and library preparation.** We sequenced 20 F1 flies individually to estimate the level of genetic variation in the founding populations (1–4 flies from each mutant pool). To prepare genomic DNA from F1 individuals, each fly was squished and incubated at 37 °C for 30 min in Squish Buffer (10 mM Tris pH 8.0, 1 mM EDTA, 25 mM NaCl, 0.15 mg/ml Proteinase K), followed by a clean-up with a Genomic DNA Clean & Concentrator kit (Zymo Research). The DNA was tagmented with a customized Tn5 protocol and sequenced in 75 bp (maximum 92 bp) paired-end on an Illumina NextSeq 500 at EMBL GeneCore.

Genomic DNA from the evolved populations was prepared using a Qiagen DNeasy Tissue Kit protocol (from Alexey Veraksa), with around 100 frozen flies (about 400 ul packed flies) per population. There are 38 samples: 18 populations × 2 generations (F3, F7) and 1 focal population (2-6-1 A) × 2 additional generations (F9, F15). They were tagmented as described above and sequenced in 50 bp (maximum 88 bp) single-end on an Illumina NextSeq 2000, with a pooling strategy intentionally biased toward higher coverage of 2-6-1 A samples.

**Read mapping and variant calling.** The reads were aligned to the dm6 genome with Bowtie2[70] (2.4.4-GCC-11.2.0), and duplicated reads were removed with Picard tools (3.1.0-Java-17). To rule out Wolbachia infection, we aligned the reads to a Wolbachia reference genome (wMelPop, GCF_00475015.1), and found 0.0 % of reads aligned in all samples. After pre-processing, we acquired a total of 89.5 million reads for the 20 F1 individuals. As a preliminary analysis, we called variants in F1 individuals with FreeBayes[71] (1.3.6-foss-2021b-R-4.1.2), with a threshold of 30 for mapping quality and 20 for base quality, on sites with a minimum coverage of 4. We found 375,779 variable positions among F1 individuals (variant quality score >10 and allele frequency <1), suggesting a substantial amount of variation in the starting populations.

For pooled-sequencing (Pool-seq) of evolved populations, we obtained an average of 5 million reads for each non-focal sample and an average of 16 million reads for 2-6-1 A samples after pre-processing. Data from F1 individuals were computationally pooled. Together our reads cover 36.6% of the genome. Despite the shallow coverage, we regard each read to be randomly sampled from the population, and the allele frequency may be roughly represented by the ratio of allele depth (AD). To extract this information, we used a pipeline adapted for Pool-seq data[34,72]: first, we realigned the reads around indels and performed base recalibration with GATK (4.2.3.0-GCCcore-11.2.0-Java-11), using the list of known variants in F1. Variable sites were then identified with bcftools mpileup and bcftools call (1.16-GCC-11.3.0), with allele depth (AD) extracted for each sample. 936,533 positions are found variable among the samples (variant quality score >10 and allele frequency <1). The variants were then annotated with ANNOVAR[73] (2020-06-08).

Unfortunately, the shallow coverage did not allow us to confidently detect EMS-induced mutations in the population data. For the non-focal populations, there were 18-56 variants private to each mutant pool (at sites with sufficient coverage), and there were 1,663 private variants for pool 2-6, which is likely associated with the high coverage on population 2-6-1 A. Therefore, we focused on common variants among the populations in the genomic analysis.

The NGS reads are deposited at ArrayExpress (EMBL-EBI) under experiment no. E-MTAB-11768.

**Estimation of EMS mutation rate.** We used the freebayes calls from the twenty F1 individuals to estimate the mutation rate induced by EMS treatment. To estimate the mutation rate, we needed to apply more stringent filters to remove background mutations. We first removed indels and sites with missing data in more than two individuals. Furthermore, we only kept sites with a mean depth between 4 and 50, and all genotypes with a depth outside this range were considered missing data. We then used bcftools +prune to remove small linkage blocks (sites with $r^2$ higher than 0.6 within a 1 kb window), which were likely to be background variation. After these filters, there were 13,292 SNPs in the dataset. We then identified SNPs that were only present in one individual (minor allele count = 1), with a requirement of at least 3 reads supporting the observed allele (AO or RO > 2). In this way, we identified 1,036 mutations across 19 mutagenized individuals (on average 55 mutations per individual) and 7 private SNPs in one non-mutagenized individual. Normalized to the number of bases covered in each

individual [generated with samtools depth (1.18-GCC-12.3.0) with the same quality and depth filter as when applying freebayes], the estimated mutation rate was on average 2.7 mutations per Mb, ranging from 0.9 to 5.4 mutations per Mb among individuals (Fig. S1c). The mutation rate was not obviously different between the two mutagenesis batches. Based on these data, we estimated the total number of novel mutations introduced to our experimental populations to be $2.7 \times 180 \, \text{Mb} \times 3500$ individuals = 1,701,000 mutations.

**Changes in allele frequency of common variants.** For each population, we used bcftools +ad-bias to apply fisher's exact test to compare allele ratio between F3 and F7, with requirements on the minimum alternative allele depth (2) and minimum depth (10). Out of the 450,739 biallelic sites tested, 54,045 (12%) sites show significant changes in allele frequency between generations in at least one population (FDR-adjusted $p < 0.05$). The changes in allele frequency span a wide range, with most changes being transitions between homozygous and heterozygous states (Fig. S3a), which is probably associated with the detection limit imposed by sequencing depth (mean depth is 29 and median depth is 21 for the sites surveyed, Fig. S3b).

Since fisher's exact test might be an overly relaxed test on allele frequency and could lead to false positives[34,74], we applied a sign test[75] to narrow down the list of variants to those showing recurrent changes in multiple populations. Each variant is given a score: $S = N_{\text{REF increase}} - N_{\text{REF decrease}}$, where $N_{\text{REF increase}}$ is the number of populations showing a significant increase in reference allele frequency and $N_{\text{REF decrease}}$ is the number of populations showing a significant decrease in reference allele frequency. Therefore, the $S$ score represents the tendency for the alternative allele to be purged (if $S > 0$) or fixed (if $S < 0$) during evolution. Out of the 450,739 biallelic sites tested, 16,394 sites (4%) showed consistent increases or decreases in allele frequency in more than one population. The mean of $S$ among these sites is 0.56, suggesting a slight systematic bias for detecting decreases in alternative allele frequency, but the majority of the changes among populations are in random directions (mean $S$ is close to 0). By using a cutoff of $S > 5$ or $S < -5$, we report on the top 1% sites (181 among 16,394) that show consistent directional changes across the parallel-evolving populations.

**Genotype-phenotype association.** Due to the low coverage and small sample size, we used genotype calls instead of allele frequency to perform genotype-phenotype association. We restricted this analysis to sites with a minimum mean depth of 10, leaving 261,167 sites in the dataset. We used the mean length of F4, F8, F10, and F17 embryos as the phenotype to associate with the 'population genotypes' of their parent generation (F3, F7, F9, and F15). Note that we used the length of F17 embryos as the phenotype of F15 population due to missing data in F16. For each variant, a linear model is used to estimate the effect size and significance of the genotype. For variants with three genotypes ("0/0", "0/1" and "1/1"), the smaller p-value is used. Due to the small sample size (30 samples at most), we don't think that the association analysis has enough statistical power to support any variant to be an interesting candidate, but the results could be used as a reference to prioritize variants detected by the sign test (e.g. the intronic G > T mutation in *CG1136* in Fig. S3e). The p-values are included in Supplementary Data 1.

**Single-nuclei transcriptomics**
2.5h-to-3.5h-old embryos (developed at room temperature) were dechorionated and flash-frozen in liquid nitrogen for nuclei preparation. The evolved embryos are from population 2-6-1 A, at the 20th generation. They were manually examined, and smaller embryos were removed upon collection to reduce noise and focus on relatively large

embryos. A wild-type line (VK33) and the 4x*bcd* lab stock were treated in parallel.

Nuclei isolation was performed following a standard protocol (10x Genomics® Single Cell Protocols, with adaptations from Francisca Hervas-Sotomayor at Heidelberg University). The frozen embryos were squished with a pestle 20 times in cold homogenization buffer (HB) [250 mM sucrose, 25 mM KCl, 5 mM MgCl$_2$, 10 mM Tris-HCl (pH 8), 0.1% Nonidet P40/IGEPAL, 1 uM DTT, 0.4 U/ul RNAse Inhibitor (New England Biolabs), 0.2 U/ul SUPERase•In™ RNase Inhibitor (Invitrogen)]. The samples were then centrifuged at 100 g for 1 min to remove unlysed tissue, and the supernatant was centrifuged at 1,000 g for 5 min to pellet the nuclei. The pellet was washed once in HB, filtered twice with Flowmi® Cell Strainers (Sigma), and resuspended in PBS. A subsample of the nuclei prep was DAPI-stained and examined under the microscope, to determine the density of nuclei. For each sample, 7,500 nuclei were used as the input for 10x library construction. RNA-seq was performed on an Illumina NextSeq 500 at EMBL Genomic Core Facilities (GeneCore) in two runs.

The reads were mapped to the *Drosophila* reference genome (dm6) plus the eGFP-Bicoid plasmid sequence and counted with Cell Ranger (6.0.1), with intronic reads included. The count data were analyzed with Seurat (3.9.9.9010)[76] in R (4.2.2), with the three samples merged into one data frame. They were first filtered to remove 1) nuclei with extremely low (<200) or high (>4,000) number of expressed genes and 2) nuclei with a high percentage of mitochondrial reads (>5%). The resulting data were normalized and scored for cell cycle status. The data were then scaled, with the percentage of mitochondrial reads, percentage of ribosomal reads, and cell cycle status regressed out. The scaled data were used for PCA, and `Harmony`[77] (0.1.1) was used to correct for batch effect with 30 PCs. A preliminary clustering was done on the corrected data with 30 PCs, and three clusters with predominantly cytosolic RNA (high percentage of ribosomal and mitochondrial RNA, low count in the number of genes and number of molecules) were removed.

After the removal, there are 3k to 6k nuclei for each sample. The data were normalized, scaled, 'harmonized', and clustered again as described above, with 30 PCs. There are 21 clusters, with no obvious cluster of doublets based on scores generated by `scrublet`[78] (0.2.3 in python 3.9.6-GCC-11.2.0). Cell types were inferred based on marker genes[79], and 11 clusters were identified as early embryonic cell types based on marker gene expression at stages 4–6 (in situ database of Berkeley Drosophila Genome Project[80]) (Table S2). Differentially expressed genes were identified with `FindMarkers` in Seurat.

To curate a set of growth-related genes to examine expression changes across cell types, we used the definition of insulin-like receptor signaling pathway in FlyBase (Gene group FBgg0000910). Other genes were curated from Choi et al.[81], Welte (2015)[39], Heier and Kühnlein[82], and Heier et al.[83].

The snRNA-seq reads are deposited at ArrayExpress (EMBL-EBI) under experiment no. E-MTAB-12068.

## Triglycerides quantification

The concentration of TGs in embryos was measured using the Triglyceride Quantification Colorimetric Kit from Sigma (Cat. #MAK266). 50 stage5 embryos were homogenized in Eppendorf tubes on a Nonidet P40 Substitute (Sigma, Cat. #74385) 5% solution. The triglycerides concentration in each homogenate was then quantified following the instructions provided by the manufacturer. Absorbance was measured at 570 nm.

## MALDI-imaging mass spectrometry on sectioned ovaries

Ovaries needed to be cryo-sectioned to prepare the tissue for MALDI-IMS. Briefly, a small number of ovaries were embedded in a previously heated 5% m/v carboxymethylcellulose (Sigma) solution. This solution then solidified at room temperature, and the resulting molds were sectioned in a Leica CM1950 cryostat at −20 °C, producing slices with a thickness of 20 μm. These slices were then mounted on regular glass slides.

The samples were then coated with a microcrystalline matrix of 2,5-dihydroxybenzoic acid dissolved in 70% acetonitrile to 15 mg/ml with the help of a TM-Sprayer robotic sprayer (HTX Technologies, Carrboro, NC, USA). The sprayer operated at a spray temperature of 80 °C, flow rate of 0.01 ml/min, track spacing of 3 mm and 10 passes, and the estimated surface concentration was 3 μg/mm^2. The glass slides were then mounted onto a custom adapter and loaded into the MS imaging ion source (AP-SMALDI5, TransMIT GmbH, Giessen, Germany). Generated ions were co-axially transferred to a high mass-resolution mass spectrometer (QExactive Plus mass spectrometer, ThermoFisher Scientific). Intact lipid imaging was performed in positive ion mode with an isolation mass range of 400–1200. Supplementary fatty acid analysis was done in negative ion mode with an isolation range of 400–1000, fragmentation energy of 45 (NCE), and product isolation between 160–320.

Metabolite annotation was performed using the METASPACE cloud software[84] with SwissLipids database[85] (version 2018-02-02). The Principal Component Analysis of these results was performed on R (4.3.2) using the FactoMineR and factoextra packages (http://factominer.free.fr/). Enrichment analyses were carried out using LION/web[86].

The metabolomics data are deposited at METASPACE.

## Dissection of ovarioles

Flies were reared in uncrowded cages with apple juice plates supplied with yeast paste for 48 h prior to dissection. 10-12 female flies were dissected for ovaries, which were kept on ice in PBT with 4% PFA until all samples were processed. The ovaries were then fixed in PBT/PFA for 30 min, washed twice in PBT, and placed in Prolong Gold with DAPI. They were then further dissected to separate the ovarioles and mounted on glass slides. The slides were imaged on a Zeiss 880 confocal microscope and scored for ovariole number and oocyte length.

## Larval behavior

Larvae (3$^{rd}$ instar, 5 days after egg laying) were harvested from food vials using a 10% glucose solution and placed on agar plates, where their movement was recorded using a FL3-U3-13Y3M-C CMOS camera (https://www.flir.de/products/flea3-usb3/) for two minutes. Then, positional information as a function of time was automatically extracted from the videos for each individual larvae using FIMtrack[87]. Behavior-related parameters (speed, bending, etc) were then calculated using this dataset.

## Reporting summary

Further information on research design is available in the Nature Portfolio Reporting Summary linked to this article.

# Data availability

The WGS and snRNA-seq reads were deposited at ArrayExpress (EMBL-EBI) under experiments E-MTAB-11768 and E-MTAB-12068, respectively. The metabolomics data were deposited at METASPACE. All data supporting the findings of this study are available within the paper and its Supplementary Information files. Source data are provided in this paper.

# Code availability

Custom R codes and source data are deposited at: https://git.embl.de/xuli/rapid-response-of-fly-populations-to-gene-dosage-across-development-and-generations.

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

## Acknowledgements

We thank Phillip Oel, Leslie Pan, Nikolaos Papadopoulos, Blanca Pijuan-Sala, and Xuefei Yuan for their help and advice in single-nuclei transcriptomics. We thank Ching-Ho Chang and Luisa Pallares for their advice on genomic analysis. We thank Dimitri Kromm and Lars Hufnagel for their help in light-sheet imaging. We thank Pinar Onal and Stephen Small for sharing the Bicoid antibody and Nicolas Gompel, Artyom Kopp, and Eileen Furlong for sharing *Drosophila* stocks. We also thank Martijn Molenaar for discussions on the interpretation of the metabolomics data. We thank Pinar Onal and members of the Crocker group for their input on the project, in particular Natalia Misunou, Mindy Liu Perkins, Albert Tsai, Rafael Galupa, Noa Ottilie Borst, and Gilberto Alvarez Canales for providing feedback on the manuscript. Mattew A. Benton, Claire Standley, and Xitong Liang also provided feedback on the manuscript. X.C.L. and L.G. are supported by fellowships from the European Molecular Biology Laboratory Interdisciplinary Postdoc Program (EIPOD) under Marie Skłodowska-Curie Actions COFUND (664726 and 847543, respectively). Research in the Crocker lab is supported by the European Molecular Biology Laboratory (EMBL).

## Author contributions

Conceptualization: X.C.L, L.G., J.C. Investigation: X.C.L., L.G., M.E., K.R., J.C. Methodology: X.C.L., L.G., M.E., T.A., J.C. Formal analysis: X.C.L., L.G., J.C. Data curation: X.C.L., L.G., J.C. Visualization: X.C.L., L.G., J.C. Software: X.C.L., L.G., J.C. Resources: T.A. Supervision: T.A., J.C. Project administration: X.C.L., J.C. Funding acquisition: J.C. Writing, original draft: X.C.L., L.G., J.C. Writing, review & editing: X.C.L., L.G., M.E., K.R, T.A., J.C.

## Funding

## Competing interests

The authors declare no competing interests.
