## [Peer Review File · Nature Communications]

Rapid response of fly populations to gene dosage across development and generationsREVIEWER COMMENTS

Reviewer #1 (Remarks to the Author):

This study uses a synthetic system to investigate how *Drosophila* may cope with an increased bicoid (*bcd*) gene dose. The authors first show that two extra copies of the *bcd* gene (4x*bcd*) can lead to a fatemap shift in the embryo and a reduced viability, largely confirming previous findings. They then mutagenize these flies in search of evolved (i.e., "mutant") lines that can ameliorate the defects. They observe a shared feature exhibited by these lines: embryo size becomes larger than the ancestral 4x*bcd* line. They then focus on one of the evolved lines (2-6-1A) for an in-depth analysis, providing evidence suggesting metabolic and developmental changes during oogenesis. While the topic of the study is timely and of interest, the actual work and its presentation do not quite match up, as detailed below.

1) This work covers a variety of potentially interesting areas. Yet it is disappointing that none of them is analyzed or presented in a satisfactory way. For example, the analysis of how the Bcd gradient input impacts the downstream gene network behavior is inadequate, particularly with respect to how the enlarged embryo may act to compensate for extra copies of *bcd*. The authors show that the gradient length scale, λ (undefined in the main text of the MS), is increased in 4x*bcd* embryos relative to wt embryos. This result appears odd and, to my best knowledge, inconsistent with the current literature (see PMID: 23580621 for example). Despite a successful detection of an expected increase in anterior Bcd intensity in 4x*bcd* embryos relative to wt embryos, there is no documentation of a quantitative nature of Bcd gradient measurements. A mechanistic interpretation of the evolution results will depend, at a minimum, on quantitative measurements of the Bcd gradient input in embryos of the wild type, 4x*bcd*, and the evolved lines.

2) A major conclusion of this work is that an increased embryo length can compensate for extra copies of the *bcd* gene. This is based on the finding that the evolved lines tend to have longer embryos than their ancestral 4x*bcd* line. It is well documented that, under normal conditions, the amplitude of the Bcd gradient has a positive correlation with embryo size (PMID: 18854140; PMID: 21613328; PMID: 25809405). In fact, the results presented in the MS show that this correlation likely also exists across species (Fig 4C). If such a positive correlation exists between the evolved lines (with larger embryos) and the ancestral 4x*bcd* line, the amplitude of the Bcd gradient in the evolved lines would be expected to be even higher, which likely would exacerbate the defects caused by 4x*bcd*, at least in the anterior part of the embryo. I did not see data on amplitude measurements in the evolved lines nor a discussion about how the (anticipated) further increase in Bcd gradient amplitude may be reconciled with the observed phenotypic relief.

3) The authors describe both transcriptomics and MALDI-imaging analyses to gain evidence about metabolic changes that may have taken place in the evolved line 2-6-1A. The motivation for such analyses comes from the fact that the evolved lines have larger embryos, which may have metabolic and/or developmental underpinnings. While this aspect of the inquiry is interesting, it is poorly presented, particularly with regard to how such changes can ultimately ameliorate the defects caused by extra copies of *bcd*. A quantitative model has been proposed to capture the *bcd* mRNA accumulation (and consequently the Bcd gradient amplitude) in relation to egg size during oogenesis (PMID: 25809405). This model may be used in the discussion and interpretation of the current results, preferably with accurate Bcd measurements that can identify Bcd-mediated effects (see PMID: 24284208 for example).

In summary, the finding that evolution can take place relatively quickly in a synthetic system is interesting. An advantage of using a synthetic system is that the input and output--in this case the Bcd gradient input and patterning output--have been well characterized, and such knowledge can help make sense of the evolution results in molecular terms. Unfortunately, the current work falls short in utilization of such knowledge when designing the experiments and interpreting the results.

Consequently, the results presented in this MS appear diffused and remain mostly at a superficial level. The short format of the submitted MS did not contribute positively to the evaluation, and it is unclear if an improved presentation alone might be sufficient to alter the assessment of this work.

Reviewer #2 (Remarks to the Author):

This paper is a study on embryonic cells that evolved through changes in the genetic environment, and used the MALDI mass spectrometry imaging as one of the methods to investigate lipid metabolism changes that occurred during this process.

In this study, the results of MALDI mass spectrometry imaging (MALDI-MSI) of cross-sectional tissue of the ovary showed that evolved embryos contained more triglycerides. Additionally, mass spectrometry results showed that fatty acids containing carbon 13-15 were more abundantly distributed in the evolved oocyte cell line.

Should changes in lipid metabolism, such as changes in triglyceride amounts and fatty acid levels, be presented through MALDI-MSI? In other words, explain the meaning that must be measured with MALDI-MSI if it could not be presented with other imaging techniques. If oocytes are from different lines - the evolved embryos and the wild type lines - wouldn't comparing them using LC-MS/MS after lipid extraction give more quantitative results?

In MALDI-MS, triglycerides are mainly measured in positive ion mode, and ionized fatty acids are mainly observed in negative ion mode because they have a negative charge. It is unclear in which ion mode the triglycerides in Figure 3(F) and the fatty acid analysis in Figure 3(G) were measured. This should be explicitly stated.

Regarding fatty acids, it was mentioned that tandem mass spectrometry (MS/MS) analysis was performed after MALDI imaging. Was MS/MS analysis conducted for all 122 lipid ions, or were only a select few ions analyzed?

The change in lipid metabolism in the evolved embryo and the need for more energy consumption were explained by the increase in the amount of triglycerides, and this interpretation is very reasonable.

If this perspective is maintained, are all the parent ions of the fatty acids measured by tandem mass spectrometry derived from triglycerides, or are there other parent ions including phospholipids? (e.g., PC, PE, PI, etc.) The reason for this question is that if the parent ions of the fatty acids presented in this study include phospholipids, then the presence of phospholipids is biologically significant, as they can participate in various cellular processes. Therefore, a decrease in the quantity of fatty acids with carbon numbers of 13 to 15 in evolved embryos may have biological implications beyond energy metabolism.

Minor:

MALDI Imaging Mass Spectrometry is commonly abbreviated as MALDI-IMS or MALDI-MSI. Apply this to the manuscript.

Line 604, Expressions such as 'MALDI-imaging mass spec' are not often used, so please replace them with the above abbreviations.

Reviewer #3 (Remarks to the Author):

I greatly enjoyed reading the manuscript by Li et al., which reports on the remarkable discovery of a system-wide compensatory effect (system size changes) to a gene network perturbation (dose increase of a network activator gene) on a relatively short evolutionary time scale (10s of generations). The authors leverage the fitness disadvantage of the synthetic network perturbation to examine the robustness and evolvability of the system. The manuscript's results point to the necessity of a systems-level view of regulatory network evolution, contrasting the typical single gene-centric reductionist approach. The work is timely and important and will be of interest to a broad readership.

Some comments that should be integrated to make the manuscript more complete and accessible:

1. Surely the authors must have thought about a similar synthetic network perturbation by halving Bicoid dosage instead of doubling it. It would be interesting to at least get a glimpse of what happens in this scenario. For example, is there also a size increase that would compensate in the anterior? Or is there a size decrease? And how would that fit with the current model proposed in the manuscript?
2. The authors should discuss their results in the larger context of system-wide network level effects of the segmentation gene network that have been reported earlier, involving 1) gap and pair-rule gene scaling with embryo size but no Bicoid scaling, and 2) gap gene boundary compensation upon Bicoid dosage challenges.
3. In Fig S2A, please explain how for some stripes there are posterior shifts when going from F4 to F8. It would be helpful to plot embryo length right next to this panel to make it clear what is going on across the many populations. It is briefly stated in the text that there is a large variability among the populations, but it would be good to discuss this in more detail.
4. Please discuss why population 2-2-2A has longer eggs, with the stripes shifting further to the posterior; something seems odd about this population. While the much-discussed population 2-6-1A has the 4x bicoid dosage leading the egg length to increase from F4 to F8, and consequently the stripe position (in relative units) moves towards the anterior (towards the wt position), for population 2-2-2A exactly the opposite seems to happen: in generation F4 they have stripes shifted back (because of 4x bcd), but by generation F8 the stripes move even further back (in rel. coordinates) despite the egg being longer (see fig S3).
5. In line 158, the size decrease is already present from Gen 8 -> 10. Could the authors comment on its significance compared to the mentioned decrease at Gen 15? Also in light of previous points, a more comprehensive statistic about the different scenarios for the different populations would be helpful; maybe with a table that lists each population analyzed and its scenario (shifts evolution, size evolution, etc...)
6. It would be useful to provide a viability score for Gen 49 (is it stable from Gen 15 on, or is it getting better/worse?)
7. In line 173 you could cite PMID 36384919 where a similar turnover has been seen in Sars-Cov2 immune escape.
8. Figure S9A - the crossing scheme would be clearer to non-Drosophila geneticists if also endogenous bicoid was annotated (could use a different color to differentiate from the transgene).
9. State somewhere in the captions of Fig1+2 that Gen 0 refers to the non-mutagenized 4xBcd, not the mutagenized one.
10. Fig 2i needs an x-axis label; otherwise unclear what is reported.

11. Large open circle data point markers in some figures (ie, 2C, 2E, 2J, 3I, 3J, 4) are difficult to parse. Please make the markers smaller and use filled circles, ie, dots...

Thomas Gregor

REVIEWER COMMENTS

Reviewer #1 (Remarks to the Author):

This study uses a synthetic system to investigate how *Drosophila* may cope with an increased bicoid (*bcd*) gene dose. The authors first show that two extra copies of the *bcd* gene (4*xbcd*) can lead to a fatemap shift in the embryo and a reduced viability, largely confirming previous findings. They then mutagenize these flies in search of evolved (i.e., "mutant") lines that can ameliorate the defects. They observe a shared feature exhibited by these lines: embryo size becomes larger than the ancestral 4*xbcd* line. They then focus on one of the evolved lines (2-6-1A) for an in-depth analysis, providing evidence suggesting metabolic and developmental changes during oogenesis. While the topic of the study is timely and of interest, the actual work and its presentation do not quite match up, as detailed below.

1) This work covers a variety of potentially interesting areas. Yet it is disappointing that none of them is analyzed or presented in a satisfactory way. For example, the analysis of how the Bcd gradient input impacts the downstream gene network behavior is inadequate, particularly with respect to how the enlarged embryo may act to compensate for extra copies of *bcd*. The authors show that the gradient length scale, λ (undefined in the main text of the MS), is increased in 4*xbcd* embryos relative to wt embryos. This result appears odd and, to my best knowledge, inconsistent with the current literature (see PMID: 23580621 for example). Despite a successful detection of an expected increase in anterior Bcd intensity in 4*xbcd* embryos relative to wt embryos, there is no documentation of a quantitative nature of Bcd gradient measurements. A mechanistic interpretation of the evolution results will depend, at a minimum, on quantitative measurements of the Bcd gradient input in embryos of the wild type, 4*xbcd*, and the evolved lines.

We agree with the reviewer that it is important to quantitatively describe the Bcd gradient, but we found that technical factors such as batch effects in fixation prevented precise quantification of Bcd concentration by immunostaining. In experimental evolution, it is, by definition, impossible to fix the evolved embryos and the pre-evolved controls at the same time. The 4*xbcd* stock often serves as a surrogate for G0 in this study but cannot represent the "real" pre-evolved population in highly sensitive quantifications. The non-mutagenized 4*xbcd* populations were fixed alongside with the mutagenized populations, but they also exhibited changes in egg length (**Fig. 1m**, 0-0-1A, 0-0-2A and 0-0-3A) and therefore cannot serve as proper controls. Finally, the observed change in egg length was transient (**Fig. S4a-b**), so it would not make sense to perform the quantification with embryos later than Gen. 10. Taken together, it was technically challenging to quantitatively compare Bcd gradients in the evolved vs. G0 embryos. We have added a paragraph of "limitation in this study" to the discussion section to explain this point (lines 322-334).

We believe that the Bcd gradient was very likely to have remained constant during experimental evolution because (1) the phenotypic readout, *eve* pattern, was still very close to the pre-evolved level, as opposed to restoring to the wild-type level (**Fig. 1I**); (2) the shape of Bcd gradient in the F10 embryos was more similar to G0 4*xbcd* embryos than 2*xbcd* embryos (**Fig. S2g**). Although the previous publication (PMID: 23580621) did not find a statistically significant difference in λ between 2*xbcd* and 4*xbcd* embryos, the study reported a 10% increase in λ as Bcd dosage *D* increases from 0.44 to 2.4 (PMID: 23580621, supplementary materials), consistent with the trend in our study. Therefore, the inconsistency might be due to technical differences in the quantifications. Because the three samples of ours were analyzed with the same technique, we believe that it is technically solid to report these results.

Although we agree that the characterization of the evolved Bcd gradient is helpful for understanding the mechanism of evolution, we think the mechanism is more likely to be upstream of Bicoid network, based on multiple lines of evidence presented in **Fig. 3**, at least for the focal population 2-6-1A. We focused on characterizing the global changes in transcriptomes and lipid metabolisms, providing a holistic view of developmental evolution that is consistent with the omnigenic model. It is possible that molecular changes in the Bicoid network occurred in other populations, which remain to be investigated in future studies.

2) A major conclusion of this work is that an increased embryo length can compensate for extra copies of the *bcd* gene. This is based on the finding that the evolved lines tend to have longer embryos than their ancestral 4*xbcd* line. It is well documented that, under normal conditions, the amplitude of the Bcd gradient has a positive correlation with embryo size (PMID: 18854140; PMID: 21613328; PMID: 25809405). In fact, the results presented in the MS show that this correlation likely also exists across species (Fig 4C). If such a positive correlation exists between the evolved lines (with larger embryos) and the ancestral 4*xbcd* line, the amplitude of the Bcd gradient in the evolved lines would be expected to be even higher, which likely would exacerbate the defects caused by 4*xbcd*, at least in the anterior part of the embryo. I did not see data on amplitude measurements in the evolved lines nor a discussion about how the (anticipated) further increase in Bcd gradient amplitude may be reconciled with the observed phenotypic relief.

Thank you for this question and the references. We noted the long-standing knowledge of the positive correlation between Bcd amplitude of embryo size and have added the references to our manuscript. However, we don't necessarily anticipate a higher Bcd amplitude in the evolved embryos, for two reasons: 1) this anticipation contradicts with recent research showing that, unlike gap genes, Bicoid does not scale (arXiv:2312.17684 [q-bio.MN]). 2) we think that there were genetic changes (either induced by EMS mutagenesis or recombined standing alleles) in the evolved lines that broke this relationship, similar to what was reported in Cheung et al. (2014) (PMID: 24284208). The positive correlation might be mediated by molecular events downstream of Bicoid, but the compensatory effects found in the present study might be mediated by upstream events, based on evidence in **Fig. 3**. For reasons described in our response to comment #1 above, we could not quantify the Bcd amplitude to directly test this hypothesis, but it is indeed interesting to take this perspective into consideration. We added discussion about Bcd and the embryonic scaling network in response to this comment and Reviewer 3's comment (lines 300-308).

3) The authors describe both transcriptomics and MALDI-imaging analyses to gain evidence about metabolic changes that may have taken place in the evolved line 2-6-1A. The motivation for such analyses comes from the fact that the evolved lines have larger embryos, which may have metabolic and/or developmental underpinnings. While this aspect of the inquiry is interesting, it is poorly presented, particularly with regard to how such changes can ultimately ameliorate the defects caused by extra copies of *bcd*. A quantitative model has been proposed to capture the *bcd* mRNA accumulation (and consequently the Bcd gradient amplitude) in relation to egg size during oogenesis (PMID: 25809405). This model may be used in the discussion and interpretation of the current results, preferably with accurate Bcd measurements that can identify Bcd-mediated effects (see PMID: 24284208 for example).

Thank you for the references and the suggestions. These are indeed very interesting papers to us, and we have added them to our manuscript. Although we agree that it is important to test hypotheses such as changes in *bcd* mRNA accumulation in relation to tissue expansion and scaling, we think it is also important to search for the molecular mechanisms unbiasedly, through genetic mapping of the causal variants or quantitative metabolic profiling, which would be of interest for future studies. Even though we originally expected direct changes in the local Bicoid network as the reviewer does, evolution surprised us: there was no evidence for molecular compensation in the local network (e.g. changes in the expression pattern of Bicoid, *hunchback*, *giant*, *Krüppel*, *knirps*; in situ data were collected but not shown due to negative results), but a change in maternal lipid metabolism. Therefore, we present our results as an example that highly interconnected genetic networks provide evolvability for previously assumed highly robust developmental systems.

We also added this point in Discussion (lines 300-308).

In summary, the finding that evolution can take place relatively quickly in a synthetic system is interesting. An advantage of using a synthetic system is that the input and output--in this case the Bcd gradient input and patterning output--have been well characterized, and such knowledge can help make sense of the evolution results in molecular terms. Unfortunately, the current work falls short in utilization of such knowledge when designing the experiments and interpreting the results. Consequently, the results presented in this MS appear diffused and remain mostly at a superficial level. The short format of the submitted MS did not contribute positively to the evaluation, and it is unclear if an improved presentation alone might be sufficient to alter the assessment of this work.

Overall, we appreciate the input from Reviewer 1 and we agree that quantification of Bicoid gradient would provide important information for understanding the mechanism of the evolved compensation, such as in the previous studies (e.g. Cheung et al. 2014). At the same time, we argue that the strength of our study is in the inter-disciplinary approach, combining evolutionary genetics, developmental biology and metabolomics, to provide a new way to think about the evolvability and robustness of developmental networks. Although technical factors prevented us from performing the quantification experiments as the reviewer suggested, we will definitely take it into account in future experimental designs. We also added a paragraph in Discussion about *bcd* scaling in response to the comments of both Reviewers 1 & 3 (lines 300-308).

Reviewer #2 (Remarks to the Author):

This paper is a study on embryonic cells that evolved through changes in the genetic environment, and used the MALDI mass spectrometry imaging as one of the methods to investigate lipid metabolism changes that occurred during this process.

In this study, the results of MALDI mass spectrometry imaging (MALDI-MSI) of cross-sectional tissue of the ovary

showed that evolved embryos contained more triglycerides. Additionally, mass spectrometry results showed that fatty acids containing carbon 13-15 were more abundantly distributed in the evolved oocyte cell line.

Should changes in lipid metabolism, such as changes in triglyceride amounts and fatty acid levels, be presented through MALDI-MSI? In other words, explain the meaning that must be measured with MALDI-MSI if it could not be presented with other imaging techniques. If oocytes are from different lines - the evolved embryos and the wild type lines - wouldn't comparing them using LC-MS/MS after lipid extraction give more quantitative results?

We agree with the reviewer that bulk MS is more sensitive than MALDI-IMS, potentially allowing the detection of subtler changes. However, the complex tissue architecture of the ovaries prevented us from using bulk methods. The aim of this assay was to test if the changes in TGs levels observed in stage 5 embryos through colorimetric tests (**Fig. 3e**) were already present in immature oocytes (inside the ovaries). Within ovaries, germ cells (oocytes, cyst cells, etc) are surrounded by somatic cells (follicle cells, polar cells, etc) which are known to have different metabolic profiles (e.g. Li, Y. et al. *Frontiers in Aging* 2 (2022): 819903). Thus, only spatially-resolved metabolomics could be used here as bulk metabolomics would not have allowed us to discern the specific cell types (oocytes or somatic cells) contributing to the metabolic differences between strains. The fact that these metabolic shifts indeed occurred in oocytes is depicted in **Fig. 3f** and semi-quantitatively in **Fig. S6b-c**. All MALDI-IMS data was normalized by a triglyceride which showed constant levels across all experiments (TG(44:3) at $m/z=767.6159$), as described in **Fig.3** legend.

We have modified the text to better reflect the aim of this assay (lines 204-209).

In MALDI-MS, tricyclerides are mainly measured in positive ion mode, and ionized fatty acids are mainly observed in negative ion mode because they have a negative charge. It is unclear in which ion mode the tricyclerides in Figure 3(F) and the fatty acid analysis in Figure 3(G) were measured. This should be explicitly stated.

Thank you for the comment. As our primary focus was on triglycerides, MALDI-IMS was carried out using the positive ion mode. **Figs. 3f** and **3g** were generated from positive mode full MS data from the same experiment. Then, based on the detected difference in FAs composition between strains we decided to perform tandem MS in negative ion mode and thus identify FAs produced by the fragmentation of heavy lipids. We have clarified the details in the protocol in Methods (lines 644-647), as well as in Results (lines 205, 214) and the legend of **Fig. 3**.

Regarding fatty acids, it was mentioned that tandem mass spectrometry (MS/MS) analysis was performed after MALDI imaging. Was MS/MS analysis conducted for all 122 lipid ions, or were only a select few ions analyzed?

The MALDI-MS/MS experiment included in **Fig. S6** was a negative mode all-ion fragmentation experiment, where a wide mass range was isolated, and fragments in the fatty acid range were measured. In other words, none of the 122 lipid ions detected in positive mode were included in the analysis. It was an independent experiment that showed a consistent trend with the MALDI-MS experiment in **Fig. 3**. We have clarified this point in Results.

The change in lipid metabolism in the evolved embryo and the need for more energy consumption were explained by the increase in the amount of triglycerides, and this interpretation is very reasonable. If this perspective is maintained, are all the parent ions of the fatty acids measured by tandem mass spectrometry derived from triglycerides, or are there other parent ions including phospholipids? (e.g., PC, PE, PI, etc.) The reason for this question is that if the parent ions of the fatty acids presented in this study include phospholipids, then the presence of phospholipids is biologically significant, as they can participate in various cellular processes. Therefore, a decrease in the quantity of fatty acids with carbon numbers of 13 to 15 in evolved embryos may have biological implications beyond energy metabolism.

Thank you for this comment. Indeed, the FAs measured by MS/MS were not derived from TGs (details are now explicitly described in the Methods section; lines 644-647). Thus, it is possible that we are detecting changes in the relative abundance of physiologically relevant phospholipids, potentially extending the biological implications of the detected metabolic alteration beyond bioenergetics. We are briefly addressing this in the discussion section now (line 337), although we think further research is needed to confirm these potential effects on cell physiology.

Minor:

MALDI Imaging Mass Spectrometry is commonly abbreviated as MALDI-IMS or MALDI-MSI. Apply this to the manuscript.

Done.

Line 604, Expressions such as 'MALDI-imaging mass spec' are not often used, so please replace them with the above abbreviations.

Done. Thank you for pointing this out!

Reviewer #3 (Remarks to the Author):

I greatly enjoyed reading the manuscript by Li et al., which reports on the remarkable discovery of a system-wide compensatory effect (system size changes) to a gene network perturbation (dose increase of a network activator gene) on a relatively short evolutionary time scale (10s of generations). The authors leverage the fitness disadvantage of the synthetic network perturbation to examine the robustness and evolvability of the system. The manuscript's results point to the necessity of a systems-level view of regulatory network evolution, contrasting the typical single gene-centric reductionist approach. The work is timely and important and will be of interest to a broad readership.

Some comments that should be integrated to make the manuscript more complete and accessible:

1. Surely the authors must have thought about a similar synthetic network perturbation by halving Bicoid dosage instead of doubling it. It would be interesting to at least get a glimpse of what happens in this scenario. For example, is there also a size increase that would compensate in the anterior? Or is there a size decrease? And how would that fit with the current model proposed in the manuscript?

Thank you for this question. Yes, we indeed thought about evolving embryos with lower dosage or hypomorphs of *bicoid*. However, the technical challenge was the severe fitness costs in these lines. In the designs of experimental evolution, there is always a trade-off between the strength of selection and "evolvability" – the stronger the selection is, the fewer individuals survive in each generation (and potentially a longer generation time), hence a slow evolution with increased genetic drifts. Therefore, we did not pursue experimental evolution with half bicoid dosage because the lines were too sick.

It would be hard to predict the consequence of selection with half dosage. The model from the present study would predict a small embryo size, but it could be one of the many possibilities. In this study, we showed that changes in embryo sizes provided the most accessible way to quickly respond to extra copies of bicoid, but the stochastic nature of evolution (occurrence of the right mutations at the right time) requires us to carefully interpret/generalize these results. We would need more evidence to predict the results of evolution in this case. We added discussion of the possibility of generalization into the manuscript (lines 327-329).

2. The authors should discuss their results in the larger context of system-wide network level effects of the segmentation gene network that have been reported earlier, involving 1) gap and pair-rule gene scaling with embryo size but no Bicoid scaling, and 2) gap gene boundary compensation upon Bicoid dosage challenges.

Thank you for this suggestion. We have added these points in Discussion (lines 301-308).

3. In Fig S2A, please explain how for some stripes there are posterior shifts when going from F4 to F8. It would be helpful to plot embryo length right next to this panel to make it clear what is going on across the many populations. It is briefly stated in the text that there is a large variability among the populations, but it would be good to discuss this in more detail.

There were indeed posterior shifts based on the heatmap of Fig. S2a. However, the posterior shifts were not statistically significant. We only found two populations with statistically significant *eve* shifts, 2-6-1A and 1-1-3A, which were described in the main text (lines 103-105). In order to clarify this point, we added the results of all the statistical tests on individual populations to **Fig. S2a-c**.

Thank you for the suggestion, we have moved the plot of embryo length (originally Fig. S3) to this figure as **Fig. S2c**.

We did not extensively discuss the diverse responses of different populations because statistically relevant interpretations were often limited by sample size when examining the parallel populations at scale. The precise measurement of *eve* stripe positions requires a strictly controlled developmental time window (mid stage 5 in this case). Consequently, we had to remove many samples in slightly earlier or later developmental stages. Because of

the very nature of experimental evolution, we cannot go back in time to fix more embryos if we found a line is interesting later. We have added the discussion of this limitation into the manuscript (lines 322-329).

4. Please discuss why population 2-2-2A has longer eggs, with the stripes shifting further to the posterior; something seems odd about this population. While the much-discussed population 2-6-1A has the 4x bicoid dosage leading the egg length to increase from F4 to F8, and consequently the stripe position (in relative units) moves towards the anterior (towards the wt position), for population 2-2-2A exactly the opposite seems to happen: in generation F4 they have stripes shifted back (because of 4x bcd), but by generation F8 the stripes move even further back (in rel. coordinates) despite the egg being longer (see fig S3).

Thank you for this sharp observation. We agree that it would be very interesting if one population showed an opposite trend to 2-6-1A, potentially suggesting a different adaptive strategy. We did not extensively analyze 2-2-2A because the sample size of F4 was very small (4 embryos, **Fig. S2c**). Although there was a significant shift in egg length for this population (**Fig. S2c**), the shifts might be caused by under-sampling of the F4 populations. The shifts in *eve* stripes in this population were not statistically significant (**Fig. S2a**), possibly also related to the small sample size. As mentioned above, we have added the discussion of the limitation in sample size into the manuscript.

5. In line 158, the size decrease is already present from Gen 8 -> 10. Could the authors comment on its significance compared to the mentioned decrease at Gen 15? Also in light of previous points, a more comprehensive statistic about the different scenarios for the different populations would be helpful; maybe with a table that lists each population analyzed and its scenario (shifts evolution, size evolution, etc...)

That is again a very sharp observation. However, the difference between Gen. 8 and Gen. 10 was not statistically significant ($p > 0.05$ by Tukey's test). We have added all pair-wise comparisons to **Fig. S4b** (former Fig. S5b) to clarify this point.

As mentioned above, we have added the results of all the relevant tests to **Fig. S2a-c**. Thanks for the suggestion!

6. It would be useful to provide a viability score for Gen 49 (is it stable from Gen 15 on, or is it getting better/worse?)

We did not have the viability score for Gen. 49 but collected the data for Gen. 35 in August 2021. We did not put the data into the manuscript because we found a batch effect in the assay: the control, 4xbcd stock, had high viability (77%) this time, inconsistent with the G0 data collected in 2019 and 2020, as well as literature results. The deviation from previous experiments could be caused by numerous environmental factors such as temperature, season, food, etc., despite a presumably constant growth condition in our fly facility. In the Gen. 35 data, we found that both the evolved line 2-6-1A and a non-mutagenized control had lower viability (61% and 71%) than 4xbcd stock (77%). Potentially, it suggests the 2-6-1A population was suffering from some long-term deleterious effects. However, due to the possible environmental changes and potentially lots of genetic drift during the thirty generations (two years, including a pandemic period when only a minimal level of maintenance work can be done with these flies), we think the viability data might not be comparable to the data collected in the first year anymore. This again highlighted the challenges of longitudinal studies with lab populations in *Drosophila*.

As mentioned above, we added a paragraph of "limitation of the study" in Discussion.

7. In line 173 you could cite PMID 36384919 where a similar turnover has been seen in Sars-Cov2 immune escape.

Thanks for the recommendation. We have added this reference.

8. Figure S9A - the crossing scheme would be clearer to non-Drosophila geneticists if also endogenous bicoid was annotated (could use a different color to differentiate from the transgene).

That is a great point. We have made the change.

9. State somewhere in the captions of Fig1+2 that Gen 0 refers to the non-mutagenized 4xBcd, not the mutagenized one.

Thank you for the suggestion. We have added this point to the legends.

10. Fig 2i needs an x-axis label; otherwise unclear what is reported.

We have made the change.

11. Large open circle data point markers in some figures (ie, 2C, 2E, 2J, 3I, 3J, 4) are difficult to parse. Please make the markers smaller and use filled circles, ie, dots...

Thank you for the suggestion. We have made the changes.

REVIEWER COMMENTS

Reviewer #1 (Remarks to the Author):

This is a revised MS. I appreciate the timeliness and the amount of this work as I expressed in the last round of evaluation. I also appreciate the technical challenges and limitations that the authors have now specifically discussed. Several issues remain that need to be addressed.

The authors' response to my previous concerns about the Bcd gradient amplitude remains unsatisfactory. There are statements in the revised MS and the response that are incorrect, or misleading at best. For example, in the revised MS (line 73), the authors attribute 4xbcd induced fatemap shift (as measured by headfold) to an "increased lambda of a steady-state exponential gradient". I do not think the authors could make such a statement based on their own results or without citing a proper reference. The authors also use lambda measurement as evidence (Fig S2g) that the evolved lines did not have a "loss of bicoid expression" (line 111). As a transcriptional activator controlling pattern formation in the embryo, it is the absolute Bcd concentration (and activity), not its lambda per se, that is responsible for downstream gene expression.

In response to my previous comment #2, the authors state that they do not anticipate a higher Bcd amplitude in the embryos of the evolved lines because, in part, Bcd gradient is known not be scaled with length. I do not think this argument addresses the issue. Indeed, it is well documented that, under normal conditions, absolute lambda (measured in absolute distance) is a property that is independent of embryo length, whereas normalized lambda (measured as fraction of embryo length) is different, with larger embryos actually having smaller relative lambda. This is due to a higher amplitude in larger embryos, and a positive correlation between amplitude and embryo size could be traced back to the amount of bcd mRNA deposited to the oocyte during oogenesis, not downstream as the authors state in their response.

The inaccurate or misleading statements in the MS (and the responses) as exemplified by those outlined above need to be corrected. At a minimum, they author should include a discussion about the possible mechanisms of how, from the perspective of Bcd gradient properties including the amplitude, the evolved lines may have ameliorated the deleterious effect caused by 4xbcd.

Another area that needs clarification is the use of chico overexpression. In the current MS, the authors merely use their result to support the argument that egg size can change in a short timescale (here with genetic manipulation). Chico plays a positive role in insulin signaling and it is unclear how overexpression of chico would result in a reduction in both egg size and triglyceride levels.

Reviewer #2 (Remarks to the Author):

The question of why triglycerides (TAGs) in evolved embryos should be measured using MALDI mass spectrometry imaging has been well articulated. Initially, the explanation for changes in fatty acids with carbon counts of 13, 14, and 15 implied a shift in lipid metabolism in the evolving embryo due to the high energy demand. However, during the revision process, it was clearly stated that these changes in fatty acids are derived not only from TAGs but also from all lipids distributed throughout the embryo, providing an indirect explanation for altered lipid metabolism in the evolving embryo. It is somewhat unclear to explain changes in lipid metabolism solely through changes in fatty acids, as changes in other phospholipids are not mentioned. However, since this issue was mentioned in the discussion section, I think there is no room for controversy about this ambiguity.

The inquiries regarding MALDI mass spectrometry imaging and other mass spectrometric results were well addressed, and if other reviewers' biochemical questions were satisfactorily answered and agreed

upon by the reviewers, I decided that the publication of the paper is feasible.

Reviewer #3 (Remarks to the Author):

I appreciate the discussion on lines 300-308 about network-level effects more broadly, but it would be more informative and forceful if it included the relevant citations that go with that statement. It seems that citations 58-63 currently included at 300-308 do not sufficiently address the network-level effects, which is necessary to highlight the connection of the results to the broader field. Perhaps refs 58 and 60 talk about the network, but only tangentially; these are primarily experimental papers.

More relevant papers on the properties of the *Drosophila* segmentation gene network to cite here are:

- 1) gap and pair-rule gene scaling with embryo size but no Bicoid scaling: Nikolic M, et al. arXiv:2312.17684 and Holloway, DM., et al. PMID: 16960857
- 2) gap gene boundary compensation upon Bicoid dosage challenges: Jaeger J, et al. PMID: 15254541
- 3) experiment & theory combinations: Dubuis J, et al. PMID: 24089448 and Petkova MD, et al. PMID: 30712870
- 4) theoretical models for the network architecture: Krotov D, et al. PMID: 24516161 and Shen, J, et al. PMID: 36546192

There is also some inconsistency in their response 2) to reviewer #1 about the evolved Bcd amplitude. As currently written, their two arguments 1) and 2) contradict each other. 1) says that Bcd does not scale, and 2) says that there were genetic differences that break the scaling of Bcd. Something needs to be justified better here. I think the authors are correct that the assumption of Bcd scaling is wrong and that counters reviewer #1's argument, but as currently stated it is unclear.

All other comments are addressed satisfactorily.

REVIEWER COMMENTS

Reviewer #1 (Remarks to the Author):

This is a revised MS. I appreciate the timeliness and the amount of this work as I expressed in the last round of evaluation. I also appreciate the technical challenges and limitations that the authors have now specifically discussed. Several issues remain that need to be addressed.

The authors' response to my previous concerns about the Bcd gradient amplitude remains unsatisfactory. There are statements in the revised MS and the response that are incorrect, or misleading at best. For example, in the revised MS (line 73), the authors attribute 4xbcd induced fatemap shift (as measured by headfold) to an "increased lambda of a steady-state exponential gradient". I do not think the authors could make such a statement based on their own results or without citing a proper reference. The authors also use lambda measurement as evidence (Fig S2g) that the evolved lines did not have a "loss of bicoid expression" (line 111). As a transcriptional activator controlling pattern formation in the embryo, it is the absolute Bcd concentration (and activity), not its lambda per se, that is responsible for downstream gene expression.

We agree with the reviewer that the lambda results only serve as indirect evidence for the properties of Bcd gradient pre- and post-evolution. We removed the lambda results (Fig. 1d and Fig. S2g) to avoid confusion; we believe the rest of the results still support our main conclusion that the compensation in eve stripe positions occurred through rapid maternal changes.

In response to my previous comment #2, the authors state that they do not anticipate a higher Bcd amplitude in the embryos of the evolved lines because, in part, Bcd gradient is known not to be scaled with length. I do not think this argument addresses the issue. Indeed, it is well documented that, under normal conditions, absolute lambda (measured in absolute distance) is a property that is independent of embryo length, whereas normalized lambda (measured as fraction of embryo length) is different, with larger embryos actually having smaller relative lambda. This is due to a higher amplitude in larger embryos, and a positive correlation between amplitude and embryo size could be traced back to the amount of bcd mRNA deposited to the oocyte during oogenesis, not downstream as the authors state in their response.

The inaccurate or misleading statements in the MS (and the responses) as exemplified by those outlined above need to be corrected. At a minimum, the author should include a discussion about the possible mechanisms of how, from the perspective of

Bcd gradient properties including the amplitude, the evolved lines may have ameliorated the deleterious effect caused by 4xbcd.

Thank you for your comment. Based on the most recent paper (Nikolic M, et al. arXiv:2312.17684), which examined Bcd concentration in 582 embryos with live imaging, we tend to agree with Reviewer #3 that the Bcd gradient does not scale, at least in highly inbred laboratory populations with a small range of egg-size variation. There is a small chance that the large embryos in our evolved lines had an altered level of Bcd protein, given that they had been mutagenized and selected. However, we are technically constrained to image the evolved and the ancestor embryos side-by-side to directly address this question.

As for the mechanism for how the large embryos ameliorate the deleterious effect of 4xbcd, we can only speculate that the enlarged posterior region relieved some of the compression caused by extra copies of bicoid. Previous studies showed that extra copies of bicoid caused a compression of fate map in abdominal regions, possibly related to lack of cell death (Namba et al. 1997 Development). Further experiments are required to determine if the large embryos in our study restored the level of cell death.

A second possibility might be that the distribution of other maternal factors, such as Nanos and Torso, changed in the large embryos and caused the compensatory shifts in *eve*, but we also need more experiments to explore this possibility.

We have added the discussion above to the manuscript (line 306-332).

Another area that needs clarification is the use of chico overexpression. In the current MS, the authors merely use their result to support the argument that egg size can change in a short timescale (here with genetic manipulation). Chico plays a positive role in insulin signaling and it is unclear how overexpression of chico would result in a reduction in both egg size and triglyceride levels.

Thank you for this comment. After careful evaluation of this result, we decided to remove it from the manuscript. Although we have repeated this experiment multiple times and we were confident that the phenotypic effects of chico expression were real, it is not central to the main message of our study. We agree with the reviewer that further experiments might be needed to explain the mechanism for the observed phenotypes, and those experiments would be outside the scope of this study. Therefore, we removed this result to keep the article clear and focused.

Reviewer #2 (Remarks to the Author):

The question of why triglycerides (TAGs) in evolved embryos should be measured using MALDI mass spectrometry imaging has been well articulated. Initially, the explanation for changes in fatty acids with carbon counts of 13, 14, and 15 implied a shift in lipid metabolism in the evolving embryo due to the high energy demand. However, during the revision process, it was clearly stated that these changes in fatty acids are derived not only from TAGs but also from all lipids distributed throughout the embryo, providing an indirect explanation for altered lipid metabolism in the evolving embryo. It is somewhat unclear to explain changes in lipid metabolism solely through changes in fatty acids, as changes in other phospholipids are not mentioned. However, since this issue was mentioned in the discussion section, I think there is no room for controversy about this ambiguity.

The inquiries regarding MALDI mass spectrometry imaging and other mass spectrometric results were well addressed, and if other reviewers' biochemical questions were satisfactorily answered and agreed upon by the reviewers, I decided that the publication of the paper is feasible.

Reviewer #3 (Remarks to the Author):

I appreciate the discussion on lines 300-308 about network-level effects more broadly, but it would be more informative and forceful if it included the relevant citations that go with that statement. It seems that citations 58-63 currently included at 300-308 do not sufficiently address the network-level effects, which is necessary to highlight the connection of the results to the broader field. Perhaps refs 58 and 60 talk about the network, but only tangentially; these are primarily experimental papers.

More relevant papers on the properties of the *Drosophila* segmentation gene network to cite here are:

- 1) gap and pair-rule gene scaling with embryo size but no Bicoid scaling: Nikolic M, et al. arXiv:2312.17684 and Holloway, DM., et al. PMID: 16960857
- 2) gap gene boundary compensation upon Bicoid dosage challenges: Jaeger J, et al. PMID: 15254541
- 3) experiment & theory combinations: Dubuis J, et al. PMID: 24089448 and Petkova MD, et al. PMID: 30712870
- 4) theoretical models for the network architecture: Krotov D, et al. PMID: 24516161 and Shen, J, et al. PMID: 36546192

There is also some inconsistency in their response 2) to reviewer #1 about the evolved Bcd amplitude. As currently written, their two arguments 1) and 2) contradict each

other. 1) says that Bcd does not scale, and 2) says that there were genetic differences that break the scaling of Bcd. Something needs to be justified better here. I think the authors are correct that the assumption of Bcd scaling is wrong and that counters reviewer #1's argument, but as currently stated it is unclear.

All other comments are addressed satisfactorily.

Thank you for the suggestions. We revised the discussion to more specifically discuss the possible mechanisms of changes in the segmentation network (line 306-332).

REVIEWERS' COMMENTS

Reviewer #1 (Remarks to the Author):

I appreciate the changes that the authors have made in response to my comments, including 1) removing the inaccurate statements about the use of Bcd gradient λ , 2) removing the chico results that lacked mechanistic details. These changes should help avoid potential confusion. The authors have also included a general discussion about how the increased egg size in the evolved lines might have ameliorated the fitness defect caused by 4xbcd. I understand the authors' reluctance in offering a more specific discussion about pre- and post-evolution Bcd gradient properties due to a lack of adequate data (but see minor issue #1 below). As commented in my initial review and acknowledged by the authors in the revised MS, their current work has raised more questions that would require future investigations. Yet overall this is an interesting study warranting a timely dissemination.

Minor issues that the authors might be able to handle on their own under the supervision of the editor.

1) line 254: "...consistent with the relationship between Bicoid and embryo size in our laboratory-evolved lines...". This statement seems to imply that the authors have data showing a higher Bcd gradient amplitude in the large embryos of the evolved lines. This needs to be clarified, by either showing the data or citing it as their own unpublished results. If this is a misstatement, then simply delete it.

2) Fig S1d: There should be two data points for Bcd4x_GO according to legend. There are three shown.

Reviewer #3 (Remarks to the Author):

The authors have adequately addressed all my comments.

Reviewer #1 (Remarks to the Author):

I appreciate the changes that the authors have made in response to my comments, including 1) removing the inaccurate statements about the use of Bcd gradient lambda, 2) removing the chico results that lacked mechanistic details. These changes should help avoid potential confusion. The authors have also included a general discussion about how the increased egg size in the evolved lines might have ameliorated the fitness defect caused by 4xbcd. I understand the authors' reluctance in offering a more specific discussion about pre- and post-evolution Bcd gradient properties due to a lack of adequate data (but see minor issue #1 below). As commented in my initial review and acknowledged by the authors in the revised MS, their current work has raised more questions that would require future investigations. Yet overall this is an interesting study warranting a timely dissemination.

Minor issues that the authors might be able to handle on their own under the supervision of the editor.

1) line 254: "...consistent with the relationship between Bicoid and embryo size in our laboratory-evolved lines...". This statement seems to imply that the authors have data showing a higher Bcd gradient amplitude in the large embryos of the evolved lines. This needs to be clarified, by either showing the data or citing it as their own unpublished results. If this is a misstatement, then simply delete it.

Thank you. We removed the misleading statement.

2) Fig S1d: There should be two data points for Bcd4x_GO according to legend. There are three shown.

Thank you for this observation. We indeed had three replicates measured over two timepoints. We have clarified it in the figure legend.

Reviewer #3 (Remarks to the Author):

The authors have adequately addressed all my comments.